

**Seasonal differences in formation processes of oxidized organic aerosol near**
**Houston, TX**
Qili Dai[1,2], Benjamin C. Schulze[2,3], Xiaohui Bi[1,2], Alexander A.T. Bui[2], Fangzhou Guo[2], Henry
W. Wallace[2,4], Nancy P. Sanchez[2], James H. Flynn[5], Barry L. Lefer[5,6], Yinchang Feng[1*], Robert
J. Griffin[2,7]
[1] State Environmental Protection Key Laboratory of Urban Ambient Air Particulate Matter Pollution
Prevention and Control, College of Environmental Science and Engineering, Nankai University, Tianjin
300350, China
[2] Department of Civil and Environmental Engineering, Rice University, Houston, TX, 77005
[3] Now at Department of Environmental Science and Engineering, California Institute of Technology,
Pasadena, CA, 91125
[4] Now at Washington State Department of Ecology, Lacey WA, 98503
[5] Department of Earth and Atmospheric Sciences, University of Houston, Houston, TX, 77004
[6] Now at Division of Tropospheric Composition, NASA, Washington, DC 20024
[7] Department of Chemical and Biomolecular Engineering, Rice University, Houston, TX, 77005
[*] Corresponding author: Yinchang Feng (fengyc@nankai.edu.cn)



**Abstract**
Submicron aerosol was measured to the southwest of Houston, Texas during winter and
summer 2014 to investigate its seasonal variability. Data from a high-resolution time-of-flight
aerosol mass spectrometer (HR-ToF-AMS) indicated that organic aerosol (OA) was the largest
component of non-refractory submicron particulate matter (NR-PM$_1$) (on average, $46 \pm 13\%$
and $55 \pm 18\%$ of the NR-PM$_1$ mass loading in winter and summer, respectively). Positive
matrix factorization (PMF) analysis of the OA mass spectra demonstrated that two classes of
oxygenated OA (less and more-oxidized OOA, LO and MO) together dominated OA mass in
summer (77%) and accounted for 42% of OA mass in winter. The fraction of LO-OOA (out of
total OOA) is higher in summer (69%) than in winter (44%). Secondary aerosols
(sulfate+nitrate+ammonium+OOA) accounted for ~76% and 89% of NR-PM$_1$ mass in winter
and summer, respectively, indicating NR-PM$_1$ mass was driven mostly by secondary aerosol
formation regardless of the season. The mass loadings and diurnal patterns of these secondary
aerosols show a clear winter/summer contrast. Organic nitrate (ON) concentrations were
estimated using the NO$_x^+$ ratio method, with an average contribution of ~15% and 37% to OA
during winter and summer campaign, respectively. The estimated ON in summer strongly
correlated with LO-OOA ($r= 0.73$) and was enhanced at nighttime.
The relative importance of aqueous-phase chemistry and photochemistry in processing
OOA was investigated by examining the relationship of aerosol liquid water content (LWC)
and the sum of ozone (O$_3$) and nitrogen dioxide (NO$_2$) (O$_x$=O$_3$+NO$_2$) with LO-OOA and
MO-OOA. The processing mechanism of LO-OOA apparently depended on relative humidity
(RH). In periods of RH <80%, aqueous-phase chemistry likely played an important role in the



formation of wintertime LO-OOA, whereas photochemistry promoted the formation of
summertime LO-OOA. For periods of high RH >80%, these effects were opposite that of low
RH periods. Both photochemistry and aqueous-phase processing appear to facilitate MO-OOA
formation except during periods of high LWC, which is likely a result of wet removal during
periods of light rain.
The nighttime increases of MO-OOA during winter and summer were 0.013 and 0.01 μg
MO-OOA per μg of LWC, respectively. The increase of LO-OOA was larger than that for
MO-OOA, with increase rates of 0.033 and 0.055 μg LO-OOA per μg of LWC at night during
winter and summer, respectively. On average, the mass concentration of LO-OOA in summer
was elevated by nearly 1.2 μg m$^{-3}$ for a ~20 μg change in LWC, which is accompanied by a 40
ppb change in $O_x$.
**1 Introduction**
Tropospheric particulate matter (PM) has adverse effects on air quality, visibility, and
ecosystems and participates in climate forcing (Watson, 2002; Grantz et al., 2003; Racherla and
Adams, 2006; Tai et al., 2010; Liu et al., 2017). The various effects of PM depend on its
physical, chemical and optical properties, which are determined by its emission, formation and
evolution/aging processes. Atmospheric PM can either be directly emitted from primary
sources (fossil fuel combustion, soil dust, sea salt, biomass burning, etc.) or formed through
chemical reactions of gaseous precursors, as is the case for secondary inorganic sulfate ($SO_4^{2-}$)
and nitrate ($NO_3^-$) and secondary organic aerosol (SOA). Understanding the source





contributions and formation pathways of PM is essential for mitigating its effects (Jimenez et
al., 2009).

Houston, TX, is of great interest to the scientific community with respect to air quality, as

it is the fourth most populous city in the United States (U.S.) and is well known for its energy
and chemical industries. The annual average National Ambient Air Quality Standard (NAAQS)
for $PM_{2.5}$ (PM with diameter smaller than 2.5 micron) set by the U.S. Environmental Protection
Agency (EPA) was recently tightened from 15 to 12 $\mu g\ m^{-3}$ (US EPA, 2013), causing Houston
to be near non-attainment of this new standard, and creating a challenging for future NAAQS
attainment (Bean et al., 2016).

Numerous efforts, from modelling (McKeen et al., 2009; Li et al., 2015; Ying et al., 2015)

to field measurements (for example, TexAQS 2000 and II (Bates et al., 2008; Parrish et al.,
2009; Atkinson et al., 2010), Go-MACCS (McKeen et al., 2009; Parrish et al., 2009),
TRAMP2006 (Mao et al., 2010; Cleveland et al., 2012), GC-ARCH (Allen and Fraser, 2006),
SHARP (Olaguer et al., 2014), and DISCOVER-AQ (Bean et al., 2016; Leong et al., 2017))
have been made in the Houston metropolitan area during the past two decades, providing
critical insights into our understanding of air quality and atmospheric chemistry with respect to
the sources and formation of PM. Previous field campaigns underscore that OA accounts for a
major fraction of non-refractory submicron PM ($NR-PM_1$) in Houston (Bates et al., 2008;
Russell et al., 2009; Cleveland et al., 2012; Brown et al., 2013; Bean et al., 2016; Leong et al.,
2017; Wallace et al., 2018). The spatial variation of $NR-PM_1$ in Houston was investigated by
Leong et al. (2017), who divided the greater Houston into two zones based on marked
differences in $NR-PM_1$ levels, characteristics, and dynamics measured at 16 sampling locations.





Zone 1 is northwest of Houston and is dominated by SOA likely driven by nighttime biogenic
organic nitrate (ON) formation. Intensive attention has been paid recently to such
anthropogenic-biogenic interactions (Bahreini et al., 2009; Bean et al., 2016). Zone 2 is the
industrial/urban area south/east of Houston. Wallace et al. (2018) found mobile source exhaust
and petrochemical emissions likely are the most important factors impacting the NR-PM$_1$ and
trace gases at a site in Zone 2.

In terms of seasonal variation, many aerosol mass spectrometer (AMS) field campaigns

worldwide have been conducted in the summer (de Gouw et al., 2008; Takegawa et al., 2009;
Lefer et al., 2010; Crippa et al., 2013a; Hayes et al., 2013; Hu et al., 2016). Intense summertime
photochemical activity in Houston was observed during TRAMP 2006 relative to other field
studies (Mao et al., 2010), indicating the potential important role of photochemical oxidation in
SOA formation in the summer (Bahreini et al., 2009). In contrast, few measurements have
focused on wintertime aerosol (Crippa et al., 2013b; Chakraborty et al., 2015; Kim et al., 2017;
Wallace et al., 2008). Wintertime aerosol generally exhibits elevated mass loadings due to the
enhanced emissions from fuel combustion for heating and weather conditions favorable to
aerosol accumulation. Only a few studies present results based on long-term measurements for
seasonal comparison, such as in the SE U.S. (Xu et al., 2015; Budisulistiorini et al., 2016). The
knowledge gap regarding aerosol seasonal variability in Houston needs to be addressed to
improve regional air quality.

Formation of SOA in clouds and the aqueous phase of aerosol particles has been reported

worldwide (Lim et al., 2010; Ervens et al., 2011; Xu et al., 2017). Given that both
photochemical oxidation and aqueous-phase chemistry are conducive to the formation of SOA,



it is of interest to compare the relative importance of photochemistry and aqueous-phase
chemistry for SOA formation in different seasons. The roles of photochemistry and
aqueous-phase processing on SOA formation and evolution in different seasons in Beijing have
been investigated by Hu et al. (2016) and Xu et al. (2017), respectively. Generally, the
aqueous-phase processing has a dominant influence on the formation of more oxidized SOA
and photochemical chemistry plays a major role in the formation of less oxidized SOA in
summer and winter in Beijing, while the relative importance of these two pathways in the
formation processes of SOA in autumn is different from those in summer and winter. The
relative roles of aqueous-phase and photochemical processes in the formation of SOA likely
vary with location and time. The seasonal differences in the spectral patterns, oxidation degrees
and contributions of SOA may result from different VOCs precursors, meteorological
conditions and atmospheric oxidizing capacity, which are not well understood in Houston,
particularly in different seasons.
This study presents observations of NR-PM$_1$ from two high-resolution time-of-flight AMS
(HR-ToF-AMS) measurement campaigns conducted during the winter and summer of 2014 at a
site in the suburbs of Houston, where industrial and vehicular emission sources and
photochemical processes are likely to play an important role in NR-PM$_1$ formation (Leong et
al., 2017). In addition to local emissions, this site was possibly impacted by regional marine
aerosol transported from the Gulf of Mexico (Schulze et al., 2018). The aims of this work are to
(1) investigate the seasonal characteristics of NR-PM$_1$ in the Houston area, (2) characterize the
primary and secondary sources by applying positive matrix factorization (PMF) analysis to the
measured OA mass spectra, and (3) evaluate the seasonal dependence of SOA composition and





formation, with a main focus on the relative effects of photochemistry and aqueous-phase
chemistry.

## 131 **2 Materials and Methods**

### 132 **2.1 Sampling Site and Campaigns**

Instrumentation was deployed in the University of Houston/Rice University Mobile Air
Quality Laboratory (MAQL), as described in Leong et al. (2017) and Wallace et al. (2018). The
winter campaign was conducted from February 3 through February 17, 2014, and the summer
campaign was conducted from May 1 to May 31, 2014. The measurement site was located on
the campus of University of Houston Sugar Land (UHSL) (29.5740 ˚N, 95.6518 ˚W). The
campus is situated southwest of downtown and the Houston Ship Channel (HSC). The map of
the measurement site is presented in Fig. S1 in the Supplemental Information (SI). The nearby
interstate highway (I-69) extends to the west of downtown and serves as a major traffic
emission source. The W.A. Parish Generating Station, a coal-fired power plant that is the
largest electricity generating facility in Texas, is ~6 miles south of the site (Fig. S1).

### 143 **2.2 Measurements**

The data used in this paper are reported in local time, which is 6 and 5 hours behind
Universal Coordinated Time (UTC) in winter and summer, respectively. The details regarding
the instrumental setup and data processing of these measurements were the same as described
in Wallace et al. (2018). The NR-PM$_1$ composition was measured using an Aerodyne
HR-ToF-AMS (DeCarlo et al., 2006; Canagaratna et al., 2007). A PM$_{2.5}$ Teflon®-coated





cyclone inlet was installed above the MAQL trailer at a height of 6 m above ground to remove
coarse particles and to introduce air into the sampling line at a rate of 16.7 SLPM. A Nafion
dryer (Perma Pure, LLC) was mounted upstream of the HR-ToF-AMS to dry the sample to
below 45% relative humidity (RH). Particles are focused into a narrow beam via an
aerodynamic lens and accelerated under high vacuum into the particle sizing measurement
chamber. After passing the particle sizing chamber, the non-refractory components are flash
vaporized at near 600$^{\circ}$C and ionized using electron impact at 70 eV. Ionized mass fragments are
then transmitted directly into the time-of-flight region so that the mass spectra can be obtained.
In this study, the HR-ToF-AMS was operated in "V-mode" to obtain the non-refractory
chemical components with a higher sensitivity, lower mass spectral resolution compared to the
"W-mode." Ionization efficiency (IE) calibration was performed monodisperse ammonium
nitrate (NH$_4$NO$_3$) at the beginning and end of each campaign. Filtered ambient air was sampled
every two days for approximately 20 to 30 min to provide a baseline of signal for the
HR-ToF-AMS during campaigns. The detection limits, (Table S1 in the SI) were calculated by
multiplying the standard deviations of the filter periods by three.

Trace gas mixing ratios and meteorological parameters also were measured on the MAQL

during the campaigns. Carbon monoxide (CO) was measured with high-resolution cavity
enhanced direct-absorption spectroscopy (Los Gatos Research, Inc.), and sulfur dioxide (SO$_2$)
was quantified using a pulsed fluorescence analyzer (ThermoFischer Scientific, model
43i-TLE). Nitric oxide (NO) and nitrogen dioxide (NO$_2$) were measured with a
chemiluminescence monitor with a UV-LED NO$_2$ photolytic converter on the NO$_2$ channel
(AQD, Inc.) The total reactive nitrogen (NO$_y$) was measured with a Thermo 49c-TL with a





heated Mo inlet converter. Ozone ($O_3$) mixing ratio was measured with ultraviolet absorption
(2BTech, Inc., model 205). Meteorological parameters including ambient temperature, solar
radiation, RH, wind speed (WS), and wind direction were measured using an RM Young
meteorological station.
**2.3 Data Processing**
The HR-ToF-AMS data analysis was performed using SQUIRREL v.1.56A and PIKA
v.1.16 in Igor Pro 6.37 (Wave Metrics Inc.). The relative ionization efficiencies (RIE) were
applied to OA (1.4), $SO_4^{2-}$ (1.2), $NO_3^-$ (1.1), $NH_4^+$ (4.0), and chloride ($Cl^-$, 1.3) following the
standard data analysis procedures. The composition-dependent collection efficiency (CE) was
applied to the data based on Middlebrook et al. (2012). Elemental ratios (H/C, O/C, N/C, and
S/C, where H is hydrogen, C is carbon, N is nitrogen, and S is sulfur) and the ratio of organic
mass to organic carbon (OM/OC) were generated using the procedures described by
Canagaratna et al. (2015).
**2.3.1 Quantification of the contributions of ON and Methanesulfonic Acid (MSA)**
*Estimation of ON.* The mass loading of $NO_3^-$ measured by HR-ToF-AMS includes both
organic and inorganic $NO_3^-$. The fragmentation ratio of $NO_2^+$ to $NO^+$ ($NO_x^+$ ratio) is different
for ON and inorganic $NO_3^-$ (Farmer et al., 2010; Fry et al., 2013), and the $NO_2^+$ and $NO^+$ mass
loadings for ON ($NO_{2,ON}$ and $NO_{ON}$) can be estimated using the method proposed by Farmer et
al. (2010):
$$NO_{2,ON} = \frac{NO_{2,obs} \times (R_{obs} - R_{NO_3NH_4})}{R_{ON} - R_{NO_3NH_4}}$$    (1)
$$NO_{ON} = NO_{2,ON} / R_{ON}$$    (2)



where $R_{obs}$ is the ambient $NO_x^+$ ratio (0.531, 0.260 for the winter and summer campaign,
respectively, see Fig. S2 for details). $R_{NO_3NH_4}$($NO_x^+$ ratio of $NH_4NO_3$) is determined by IE
calibration using monodisperse $NH_4NO_3$ before and after the campaigns. The average of the
two IE calibrations was used as the $R_{NO_3NH_4}$ for the campaign (0.588, 0.381 for the winter and
summer campaigns, respectively), which is comparable with the value reported elsewhere (Xu
et al., 2015; Zhu et al., 2016). The value of $R_{ON}$ is hard to determine because it varies with
instruments and precursor volatile organic compounds (VOCs) (Fry et al., 2013). As
summarized by Xu et al. (2015), $R_{ON}$ values ranging from 0.1 to 2.0 likely correspond to the
upper and lower bounds of the ON concentration estimated by the $NO_x^+$ ratio method. In this
work, $R_{ON}$ is adopted as 0.166 as reported in literature (Fry et al., 2009). In winter, $R_{obs}$ was
significantly higher than $R_{ON}$ and close to $R_{NO_3NH_4}$, implying significant existence of
inorganic $NO_3^-$. In summer, $R_{obs}$ was lower than $R_{NO_3NH_4}$ and close to $R_{ON}$, indicating a
significant fraction of the total $NO_3^-$ is ON (Fig. S2).

The measured $NO_x^+$ ratio can be used to separately quantify ammonium and organic

nitrates as:
$ON_{frac} = \frac{(R_{obs}-R_{NO3NH4})(1+R_{ON})}{(R_{ON}-R_{NO3NH4})(1+R_{obs})}$                    (3)
The nitrate functionality from organic nitrate was calculated as:
$NO_{3,ON} = ON_{frac} \times NO_3^-$                    (4)
Thus, the nitrate functionality from inorganic nitrate (assuming $NH_4NO_3$ is the solely important
inorganic nitrate in the submicron mode) can be calculated as:
$NO_{3,AN} = \left(1 - ON_{frac}\right) \times NO_3^-$                    (5)



The estimation of the total mass of ON via this method is uncertain as the actual molecular
weight of the particle-phase species is unclear. Generally, the mass of ON is estimated by
assuming that the average molecular weights of organic molecules with nitrate functional
groups (value determined as described above) are 200 to 300 g mol$^{-1}$ (Surratt et al., 2008;
Rollins et al., 2012). Previous work found that the nitrate radical (denoted as NO$_3$· with a dot to
differentiate it from aerosol NO$_3^-$) reaction with monoterpenes resulted in significant SOA
formation and that a hydroperoxy nitrate (C$_{10}$H$_{17}$NO$_5$) was likely a major NO$_3$·-oxidized
terpene product in the southeastern U.S. (Ayres et al., 2015). Here, we use the molecular
weight of C$_{10}$H$_{17}$NO$_5$ (231 g mol$^{-1}$) to calculate the ON mass. Example periods of significant
ON contribution to PM are given in Fig. S3.
*Estimation of MSA*. During the two campaigns, there is no significant organic sulfur
contribution from other ion fragments except for CH$_3$SO$_2^+$. The concentration of MSA was
estimated as:
$C_{MSA} = \dfrac{C_{CH3SO2}}{f_{MSA,CH3SO2}}$                                        (6)
where $C_{CH3SO2}$ is the concentration of ion fragment CH$_3$SO$_2^+$ (*m/z*=78.99) and the fraction of
CH$_3$SO$_2^+$ to the total signal intensity of all the fragments of pure MSA, $f_{MSA,CH3SO2}$, is 5.55%.
This values was observed for the mass spectra of pure MSA in laboratory experiments (Schulze
et al., 2018) and is comparable to previous work (Huang et al., 2015).
**2.3.2 Positive Matrix Factorization (PMF) Analysis**
The PMF technique has been used widely for source apportionment (Paatero and Tapper,
1994), including with HR-TOF-AMS data (Ulbrich et al., 2009; Zhang et al., 2011). The



high-resolution NR-PM$_1$ OA mass spectra matrix (m/z =12 to m/z=130) and the associated
error matrix obtained by using PIKA v 1.19 D were used for PMF analysis. Data were prepared
according to the protocol proposed by Ulbrich et al. (2009) and Zhang et al. (2011) prior to
PMF analysis. The PMF model was used to decompose the measured OA mass spectra matrix
by solving:
$X = GF + E = \sum_{p=1}^{J} G_{ip}F_{pj} + E_{ij}$ (7)
where X is the $m \times n$ matrix of measurement data, the $m$ rows of X are the OA mass spectra
measured at each time step, the $n$ columns of X are the time series of each organic
mass-to-charge ratio, and $p$ is the number of factors. $G_{ip}$ is the matrix that denotes the
contributions of factor $p$ at time step $i$, and $F_{pj}$ represents the factor mass spectral profiles. E
is the residual matrix. The least-squares algorithm is used to fit the data to minimize iteratively
a quality of fit parameter, Q:
$Q = \sum_I \sum_J (E_{ij}/\sigma_{ij})^2$ (8)
where $\sigma_{ij}$ is the matrix of estimated errors of the data.
Solutions using PMF with 2 to 7 factors were explored. The best solution with the
optimum number of factors was evaluated carefully using an open source PMF evaluation tool
(PET v 2.08D, (Ulbrich et al., 2009)) following the procedures described in Zhang et al. (2011).
Selection criteria included 1.) variation of the ratio of Q to expected Q$_{exp}$ (*mn−p(m+n)*, the
degrees of freedom of the fitted data (Paatero et al., 2002)) after adding an additional factor, 2.)
agreement between the reconstructed OA mass concentrations and the measured concentrations,
3.) scaled residuals for the different ion fragments included in the dataset and variations of the





residual of the solution as a function of time, 4.) agreement between factor time series and time
series of external tracers/individual ions, and 5.) examination of factor profiles. The last two are
considered to determine the physical meaningfulness of the factors. The PMF solution with
factor numbers greater than five and four for winter and summer dataset, respectively, yielded
no new distinct and physical meaningful factors. The $Q/Q_{exp}$ and the factors obtained for
different FPEAK (from -1 to 1 with a step value of 0.2) values resulted in a small difference in
the OA components. Because of the lowest $Q/Q_{exp}$ and because the use of FPEAK values
different from 0 did not improve the correlations between PMF factors and potentially
associated tracers, the five- and four-factors solutions with FPEAK=0 can be well interpreted in
winter and summer, respectively. The convergence of the PMF model containing five- and
four-factors were examined by running each model from fifteen different starting values
(SEEDs 0-30 with a step value of 2). The small variation observed in $Q/Q_{exp}$ and the mass
fraction of different factors as SEED changed indicates the solutions were stable. As a result,
SEED 0 was chosen for the final solution. The factors were interpreted as hydrocarbon-like OA
(HOA), biomass burning OA (BBOA), cooking OA (COA, identified only in the winter
campaign), and two oxidized OA (named less-oxygenated (LO-) OOA and more-oxygenated
(MO-) OOA). The data treatment, factor selection and interpretation are detailed in the SI.
**2.3.3 Estimation of Aerosol Liquid Water Content (LWC)**
Aerosol LWC includes water associated with organic aerosol and inorganic aerosol, which
were calculated using an empirical method and a thermodynamic model, respectively.
Inorganic LWC ($W_i$) was predicted by ISORROPIA-II in forward mode in mol L$^{-1}$ (Fountoukis
and Nenes, 2007). Inputs for ISORROPIA-II include inorganic aerosol mass concentrations





($SO_4^{2-}$, inorganic $NO_3^-$, and ammonium ($NH_4^+$)) and meteorological parameters (temperature
and RH). Calculation empirical of organic LWC ($W_O$) follows (Petters and Kreidenweis, 2007;
Guo et al., 2015):
$W_O = \frac{m_{org}\rho_w}{\rho_{org}} \frac{\kappa_{org}}{(1/RH - 1)}$                    (9)
where $m_{org}$ is the organic mass concentration (µg m$^{-3}$), and $\rho_w$ is the density of water (1 g
cm$^{-3}$). The organic density ($\rho_{org}$, g cm$^{-3}$) was estimated using an empirical equation based on
elemental ratios (Kuwata et al., 2012; Guo et al., 2015):
$\rho_{org} = 1000 \times [\frac{12 + \frac{H}{C} + 16 \times \frac{O}{C}}{7.0 + 5 \times \frac{H}{C} + 4.15 \times \frac{O}{C}}]$                    (10)
The hygroscopicity of SOA generated during chamber studies under sub-saturated regimes
depends on the OA degree of oxidation (Prenni et al., 2007; Jimenez et al., 2009; Petters et al.,
2009; Chang et al., 2010). A simple linear relationship between the OA degree of oxidation
(defined as the fraction of the total signal at m/z 44, f$_{44}$) and hygroscopicity ($\kappa_{org}$) is used
(Duplissy et al., 2011):
$\kappa_{org} = 2.2 \times f_{44} - 0.13$                    (11)
The total LWC is then found by summing the water content associated with each mass fraction:
$LWC = W_i + W_O$                    (12)

**3 Results and Discussion**
**3.1 Temporal Dependences of Submicron Aerosol Composition**
Campaign overview data for winter and summer are shown in Fig. 1, including



meteorological parameters (e.g., temperature, RH, radiometer, wind direction and speed), trace
gases (e.g., CO, $SO_2$, NO, $NO_2$, and $O_3$), chemically resolved NR-$PM_1$ concentrations, OM/OC,
and elemental ratios (H/C, O/C, N/C and S/C). Data also are shown in Table 1.

Data indicate that the average concentration of NR-$PM_1$ during winter campaign was 6.0 ±

3.7 µg m$^{-3}$, ranging from 0.5 to 14.8 µg m$^{-3}$. Mass loadings of NR-$PM_1$ at this measurement site
are relatively smaller than a site near the HSC in winter 2015 (10.8 µg m$^{-3}$ (Wallace et al.,
2018)), perhaps suggesting a weaker industrial influence at the UHSL site.

The average concentration of NR-$PM_1$ during summer was 3.6 ± 1.7 µg m$^{-3}$, ranging from

0.3 to 13.7 µg m$^{-3}$. For comparison, a summer campaign in 2006 on an elevated building near
downtown Houston showed an average NR-$PM_1$ concentration of approximately 11 µg m$^{-3}$
(Cleveland et al., 2012). An elevated NR-$PM_1$ episode was observed from May 28-31 (Fig.
1(m)), with high solar radiation and $O_x$ ($O_x$ = $NO_2$ + $O_3$) levels during the daytime, and high
RH at night, resulting in OA becoming the largest fractional species, likely due to gas-phase
photochemical production of SOA together with the nighttime increase of SOA associated with
high RH, lowered boundary layer and cooler temperatures.

In winter, OA was the largest component of NR-$PM_1$, accounting for 45.5 ± 13.3% on

average of the total mass, followed by $SO_4^{2-}$ (19.9 ± 11.2%), $NO_{3,AN}^-$ (17.2 ± 10.8%), $NH_4^+$
(13.2 ± 5.4%), $NO_{3, ON}$ (3.4 ± 1.4% ) and $Cl^-$ (0.9 ± 0.2%) (Fig. 2). Primary OA
(POA=HOA+BBOA+COA) was responsible for 59.1 ± 19.2% of OA mass. Secondary species
($SO_4^{2-}$+$NO_3^-$+$NH_4^+$+LO-OOA+MO-OOA) accounted for ~72.3 ± 18.1% of NR-$PM_1$ mass,
which is higher than that in winter in Seoul (Kim et al., 2017) and Beijing (Hu et al., 2016).
The inorganic aerosols in the winter were mostly neutralized in the forms of $NH_4^+$ salts (e.g.,



$(NH_4)_2SO_4$, $NH_4NO_3$, $NH_4Cl$) based on the predicted-to-measured $NH_4^+$ ratio of ~1 with
correlation coefficient ($r^2$) of 0.98 (Fig. 3(A)).

In contrast to winter, OA during the summer campaign constituted on average 54.6 ± 18.2%

of NR-PM$_1$ mass, and $SO_4^{2-}$ was the second largest component (30.9 ± 15.5%), followed by
$NH_4^+$ (12.2 ± 5.2%). $NO_{3,ON}^-$ and $NO_{3,\,AN}$ only accounted for 1.5 ± 1.9% and 0.4 ± 0.8% of
NR-PM$_1$ mass in the summer, respectively. Cl$^-$ contributed 0.4 ± 0.5% of NR-PM$_1$ mass. The
increased PBL height in summer (Haman et al., 2012) likely contributed to relatively lower
trace gas and NR-PM$_1$ levels in the summer. Secondary species contributed ~87.3 ± 14.2% of
NR-PM$_1$ mass, indicating that the relative importance of secondary aerosol formation increased
during summer as compared to winter, especially for species such as $SO_4^{2-}$ and MO-OOA.

The total OA displayed high values during the nighttime hours in both winter and summer,

maintaining a high level until morning rush hour, and then decreasing to a minimum value after
9:00 (Fig. 4). The summertime OA presented a small peak at noon, suggesting that
photochemical formation of OA played a more important role in summer than in winter.
Increasing of ambient temperature and PBL height after sunrise causes re-partitioning to the gas
phase, likely contributing to the decrease of OA, LO-OOA and ON during daytime.

Contributions of PMF factors to wintertime and summertime OA show significant

differences. For wintertime OA, on average, BBOA, MO-OOA, and COA made similar
contributions of 24%, 23% and 22% to total OA mass, respectively. The LO-OOA accounted
for 18% of OA mass, followed by HOA (13%). The POA constituted more than half of OA
mass (59%), with the remainder of being OOA (41%). In the summer, LO-OOA represented the
largest fraction of the OA mass (53% on average), followed by MO-OOA (24%), HOA (12%)





and BBOA (11%). In the case of summer, OOA constituted 77% of OA and 42% of total
NR-PM$_1$ mass, which are almost two times their relative contributions in winter. The time
series of mass concentrations of NR-PM$_1$ species (Fig. 1) and OA factors (Fig. 5) in summer
were relatively stable and repeatable, while it changed dramatically in winter due to the
different meteorological conditions.
**3.2 Seasonal Variation of the Formation of Sulfate and Nitrate**
During the summer campaign, the prevailing southerly winds from the Gulf of Mexico
carry marine aerosols to Houston (Schulze et al., 2018), resulting in a relatively high fraction of
SO$_4^{2-}$ and MSA. As shown in Fig. 1(m, j), the increased contribution of SO$_4^{2-}$ occurred when
winds originated from the south at a high speed (e.g., May 16-27), while the contribution of
SO$_4^{2-}$ decreased significantly when winds originated from the north (e.g., May 10$^{th}$ and May
13-15). MSA and S/C were markedly elevated during periods of southerly winds (Fig. 1(o, p)),
and O/C and OM/OC were relatively higher (Fig. 1(n)). In addition, elevated SO$_2$ plumes were
recorded during periods of southerly winds (Fig. 1(j, k)), potentially as a result of emissions
from the Parish coal-fired power plant. In contrast to SO$_4^{2-}$, the fractional contribution of NO$_3$
and OA increased greatly when the winds were not southerly. Primary pollutants such as CO,
NO and NO$_2$, were elevated when winds were northerly (Fig. 1(k)), accompanied by lower O/C
and higher H/C ratios during the corresponding periods (Fig. 1(n), e.g., May 1$^{st}$, 2$^{rd}$, 10$^{th}$, 15$^{th}$).
Diurnal patterns of NR-PM$_1$ and other species in the winter and summer (Fig. 4) suggest
significant seasonal dependence of sources and formation processes of NR-PM$_1$ species in
Houston. In the case of SO$_4^{2-}$, the diurnal pattern displayed a daytime peak in both winter and
summer, with the peak much more pronounced in summer mid-day. In winter, the $f_{SO4}$ (mole



ratio of [$SO_4^{2-}$] to the sum of [$SO_2$] and [$SO_4^{2-}$]) and LWC have concurrent peak value during
the night time. However, there is no obvious correlation between $f_{SO4}$ and LWC in summer,
though a moderate correlation ($r = 0.44$) was found in winter (Fig. 3). These results suggest that
$SO_4^{2-}$ formed though aqueous-phase chemistry in winter is more prominent than that in
summer.

The total nitrate concentration was higher in winter than in summer. $NO_3^-{}_{,AN}$ was very low

in summer due to its thermal instability under high temperature, while it was relatively
enhanced in winter. According to the $NO_x^+$ ratio method described in Sec. 2.3.1, the mass
fraction of $NO_3^-{}_{,AN}$ in total nitrate was decreased from 90% (1.26 µg m$^{-3}$) in winter to 48%
(0.04 µg m$^{-3}$) in summer. The concentration of $NO_{3,ON}$ was 0.14 µg m$^{-3}$ in winter, which is 3.5
times that in summer. The seasonal variation of $NO_3^-{}_{,AN}$ is much stronger than that of $NO_{3,ON}$.
This is in accordance with previous observations in Atlanta, Georgia and Centreville, Alabama
(Xu et al., 2015).

The diurnal profiles of $NO_{3,ON}$ show that it reached peak value before dawn in both seasons

(Fig. 6). However, $NO_3^-{}_{,AN}$ presents a bimodal diurnal profile in both seasons. The $NO_3^-{}_{,AN}$,
which increased from late afternoon and peaked at 2:00-4:00, was likely formed through
nighttime chemistry from dinitrogen pentoxide ($N_2O_5$) hydrolysis, as the LWC displayed a
trend similar to that of $NO_3^-{}_{,AN}$, This was corroborated by the observation of $O_x$ (>25 ppb),
which is needed to form $N_2O_5$ (via $NO_3^·$). The second peak observed during morning rush hour
was likely formed though photochemical processing of $NO_x$ emitted from vehicles because the
traffic flow and $O_x$ level are elevated during morning rush hour. The decreasing trend of
$NO_3^-{}_{,AN}$ after 9:00 is presumed to be a result of enhanced PBL height and evaporation.



The estimated ON accounted for ~1 to 57% of the total NR-PM$_1$ and 1 to 99% percent of
the OA with an average contribution of about 12 and 37% to both, comparable to other studies
(Fry et al., 2009; Rollins et al., 2010; Xu et al., 2015; Berkemeier et al., 2016). In winter, ON,
on average accounted for 35 and 15% of NR-PM$_1$ and OA mass, respectively. Figure S3
presents a high ON loading period observed in summer.
A proxy for NO$_3\dot{}$ production rate is based on the product of the observations of [NO$_2$] and
[O$_3$] (Rollins et al., 2012), where brackets represent mixing ratios in ppb. The O$_x$ (> 25 ppb)
and elevated NO$_x$ observed at night in summer (Fig. 4) resulted in rapid NO$_3\dot{}$ formation. Thus,
the concurrent enhancement in ON and O$_3$ times NO$_2$ occurring during nighttime (Fig. S3)
presumably was caused by the nocturnal NO$_3\dot{}$-initiated oxidation of anthropogenic and
biogenic VOCs, with the latter probably larger than the former (Brown et al., 2013). The high
N/C ratio of LO-OOA, concurrent peak value in LO-OOA and ON during nighttime hours (Fig.
4), and appreciable correlation of LO-OOA and ON in summer ($r$ = 0.73) (Fig. 5) together
suggest that particle-phase ON from NO$_3\dot{}$-initiated chemistry contributed to nighttime
LO-OOA in summer.

**3.3 Effects of Aqueous-phase and Photochemical Oxidation on OOA Formation**
On average, OOA accounted for 41 ± 19% of OA mass in winter but increased to 77 ± 16%
in summer. Note that MO-OOA accounted for more than half of OOA in winter (56%),
indicating the more important role of MO-OOA in winter as compared to LO-OOA on a
relative basis. In contrast, LO-OOA dominated OOA in summer (69%). The mass spectra of
MO-OOA in winter and summer are similar (Fig. 7, $r$ = 0.84) as are the extent of oxidation





(O/C = 1.10 versus 1.07). However, LO-OOA in winter showed a different spectral pattern
compared with that in summer. The mass spectrum of LO-OOA in winter was characterized by
high $m/z$ 32 (mainly $CH_4O^+$) and 46 (mainly $CH_2O_2^+$) peaks, resulting in a relatively high O/C
(0.89) in winter that suggest LO-OOA in winter was more aged than that in summer
(O/C=0.74).
Sun et al. (2016) reported a unique OOA in ambient air, termed aq-OOA
(aqueous-phase-processed SOA), that strongly correlated with particle LWC, sulfate and
S-containing ions. As shown in Table 2, by comparing the mass spectra of OOA in this work
with aq-OOA, it is found that the mass spectra of MO-OOA in winter in this study presents a
much stronger correlation ($r$ =0.96) with aq-OOA, rather than LO-OOA in winter in this study
($r$ =0.75). Both MO-OOA and LO-OOA in summer highly correlated with aq-OOA. This result
indicates that the formation of LO-OOA in summer and MO-OOA in both seasons may involve
aqueous-phase chemistry.
Assuming that OOA deduced from PMF analysis can be used as a surrogate of SOA (Wood
et al., 2010; Xu et al., 2017), the two OOA were used to investigate the formation mechanisms
and evolutionary processes of SOA. Previous studies have found SOA correlated well with odd
oxygen ($O_x$) in many cities (Wood et al., 2010; Sun et al., 2011; Hayes et al., 2013; Zhang et al.,
2015; Xu et al., 2017) and that SOA formation is significantly impacted by aqueous-phase
processing (Lim et al., 2010; Ervens et al., 2011; Xu et al., 2017). The relationships between
OOA factors and $O_x$/LWC were used as the metrics to characterize SOA formation mechanisms
associated with photochemistry/aqueous oxidation chemistry (Xu et al., 2017).
Fig. 8 (A, B) indicates the LWC frequency distribution. Winter LWC are binned in 5 μg





m$^{-3}$ increments from 0 to 20 μg m$^{-3}$. Data in the ranges of 20 to 30 μg m$^{-3}$, 30 to 50 μg m$^{-3}$, 50
to 80 μg m$^{-3}$, and 80 to 120 μg m$^{-3}$ are shown as 25, 40, 65 and 100 μg m$^{-3}$, respectively.
Summer LWC are binned in 2.5 μg m$^{-3}$ increments from 0 to 15 μg m$^{-3}$. The bins shown as 17.5
and 27.5 μg m$^{-3}$ represent data from 15 to 20 μg m$^{-3}$ and 20 to 35 μg m$^{-3}$.
The data associated with the artificially created bins in both seasons did not pass the
normal test and homogeneity test of variances. The statistical significance of differences
between bins was then tested using the Kruskal-Wallis analysis of variance (K-W ANOVA).
The differences between winter and summer data of the bins were significant. Thus, the
Dunn-Bonferroni test was performed for the *post-hoc* pairwise comparisons. It was found that
the difference of all measured variables in different bins shown in Fig. 8 were significant
($p<0.01$). The results can be found in Tables S5-S6. Fig. 8(C, D) presents a clear positive trend
of RH as a function of LWC in both winter and summer which implies an increased potential
for aqueous-phase processing at high RH level, enhanced by low wind speed that allows
accumulation of pollutants (Fig. 8(E, F)). The patterns of other parameters as LWC increases in
winter were different from those in summer.
The variation of binned mean OA mass against LWC presents significant seasonal
difference (Fig. 8(A, B)). In winter, the OA mass increased when LWC increased from 2.5 to
12.5 μg m$^{-3}$ but decreased as the LWC increased further. The LO-OOA mass decreased
dramatically when LWC>12.5 μg m$^{-3}$ (RH>80%, Fig. 8(C)) while MO-OOA continues
increasing until LWC> 40 μg m$^{-3}$. This result indicates that wet removal may dominate under
an extremely high RH environment coupled with stagnant air (WS <2 m/s Fig. 8(E)). In
summer, the OA mass decreased when LWC increased from 1.25 to 6.25 μg m$^{-3}$ but increased





when LWC increased further, suggesting the wet removal effect is not as strong as that in
winter because of the relatively lower LWC in summer than in winter.
On average, LO-OOA (Fig. 8(G, H)) in winter increased from 0.3 to 0.9 µg m$^{-3}$ when LWC
increased from 2.5 to 7.5 µg m$^{-3}$ but decreased as the LWC increased further, particularly when
LWC >40 µg m$^{-3}$. The slope of this decrease was approximately -0.008 µg LO-OOA µg$^{-1}$ LWC.
Fig. 8(A) shows that 64% of the data points were observed in the situation of low LWC (<12.5
µg m$^{-3}$, RH<80%), when the increase of LO-OOA was more significant than that of MO-OOA.
In contrast, LO-OOA in summer showed a decreased trend under low LWC level (LWC<6.25
µg m$^{-3}$, RH<80%) but a significant linear increase from approximately 0.77 µm$^{-3}$ to 1.8 µg m$^{-3}$
as LWC increased from 6.25 to 27.5 µg m$^{-3}$, a slope of 0.053 µg LO-OOA µg$^{-1}$ LWC. The
relatively high LO-OOA under low LWC level was likely more regional, with contributions
from possibly transported non-aqueous OOA, as the wind speed in this case was relatively high
and RH was low. The formation of LO-OOA under high LWC level was likely enhanced by
local aqueous-phase heterogeneous chemistry.
MO-OOA (Fig. 8(I, J)) slightly increased during both seasons as LWC increased. In winter,
MO-OOA presented a similar linear increase trend from 0.57 to 0.98 µg m$^{-3}$ when LWC
increased from 2.5 to 40 µg m$^{-3}$ but decreased as the LWC increased further (probably due to
the wet removal effect). The slope of this increase was approximately 0.008 µg MO-OOA µg$^{-1}$
LWC. In summer, MO-OOA appears to increase from 0.49 to 0.64 µg m$^{-3}$ when LWC increased
from 2.5 to 27.5 µg m$^{-3}$, with slope of 0.005 µg MO-OOA µg$^{-1}$ LWC. In winter, because of the
decrease in LO-OOA with LWC, the relative fraction of MO-OOA increases as LWC increases.
The mutual effect of aqueous-phase and photochemistry on OOA formation prevents solely



evaluating the role of the two processes. Sullivan et al. (2016) reported multiple lines of
evidence for local aq-SOA formation observed in the Po Valley, Italy during times of increasing
RH, which coincided with dark conditions. Thus, the daytime data were separated to examine
the variation of OOA against $O_x$. The relationship between OOA and aqueous-phase chemistry
was investigated further by excluding the daytime data, with the aim of diminishing the
influence of photochemistry. To do so, nighttime and daytime were based on sunrise and sunset
in Houston during the two campaigns (https://www.timeanddate.com/sun/usa/houston). On
average, the day lengths are 11 h 10 min and 13 h 35 min for the campaigns in February and
May, 2014, respectively.
Figure 9 presents the scatter plots of OOA versus LWC during nighttime for the two
campaigns. The green dots denote the increasing trend of OOA against LWC. It is found that
the increase of wintertime LO-OOA under low LWC level (<20 µg m$^{-3}$) during the night is
stronger than that shown in Fig. 8 (G). The nighttime LO-OOA linearly increased from 0.04 to
0.64 µg m$^{-3}$ when LWC increased from 2.5 to 17.5 µg m$^{-3}$, a slope of 0.033 µg LO-OOA µg$^{-1}$
LWC. This result indicates that the nighttime increase in LO-OOA in winter is more likely
formed via aqueous-phase chemistry in aerosol liquid water. In contrast, the increase of
LO-OOA under high LWC level (LWC>6.25 µg m$^{-3}$) in summer was less enhanced during
nighttime (0.055 µg LO-OOA µg$^{-1}$ LWC) as compared to the increase rate of whole dataset
(0.053 µg LO-OOA µg$^{-1}$ LWC). The slope of nighttime increase of MO-OOA against LWC
during the winter campaign was 0.013 µg MO-OOA µg$^{-1}$ LWC, which is 1.7 times the slope for
the whole dataset (daytime and nighttime). For the summer campaign, the increase of nighttime
MO-OOA is 2.2 times the rate for the whole dataset.



These results suggest that aqueous-phase processing likely has a strong positive impact on
the formation of MO-OOA in the two seasons except for instances when LWC exceeds 100 µg
m$^{-3}$ in winter. It also appears to facilitate the local formation of LO-OOA under low LWC level
(<17.5 µg m$^{-3}$) in winter and under relatively high LWC level (>6.25 µg m$^{-3}$) in summer.
As mentioned previously, ON contributes significantly to summertime LO-OOA, and the
concurrent enhancement in ON and LO-OOA during night was associated with elevated RH
(Fig. 4). A previous study found that the partitioning of organic compounds to the particle phase
was significantly increased at elevated RH levels (70%) in an urban area dominated by
biogenic emissions in Atlanta (Hennigan et al., 2008). The correlation of ON and LO-OOA in
summer nighttime ($r$=0.76) was stronger than that during daytime ($r$ =0.53). Thus, we presume
that aerosol water facilitates the formation of ON from $NO_3\cdot$-initiated chemistry involving
BVOCs during nighttime, resulting in a good relationship of LO-OOA and LWC in summer.
MSA is a secondary product from the oxidation of dimethyl sulfide (DMS) (Zorn et al.,
2008), which is a gaseous species emission from marine organisms (Barnes et al., 2006). Thus,
MSA is found to be abundant in marine/coastal areas and play an important role in the
formation of marine PM (Gondwe, et al., 2004; Huang et al., 2015; Schulze et al., 2018). The
formation of MSA is unique to aqueous-phase processing, and could be used as an indicator of
aqueous SOA formation (Barnes et al., 2006; Ervens et al., 2011). Recent observations
confirmed that MSA and associated fragment ions ($CH_2O_2^+$ (m/z 46), $C_2O_2^+$ (m/z 56) and
$C_2H_2O_2^+$ (m/z 58), which are unique ions of glyoxal and methylglyoxal uptake on SOA
(Chhabra et al., 2010)) strongly correlated with SOA formed via aqueous-phase processing (Ge
et al., 2012; Sun et al., 2016). In this work, the MO-OOA formation was associated with





aqueous-phase oxidation more strongly than LO-OOA in winter, which likely can be further
verified by the correlations between MO-OOA/LO-OOA and MSA. As shown in Fig. 7, MSA
has a relatively higher correlation coefficient with MO-OOA ($r$=0.45) compared to LO-OOA
($r$=0.30), though the correlation also is influenced by many other factors.

Fig. 10 (A, B) presents the frequency distribution of $O_x$. Winter $O_x$ are binned in 10 ppb

increments from 0 to 60 ppb. The range for summer is 20 to 70 ppb. The data associated with
the artificially created $O_x$ bins in both seasons did not pass the normal test and homogeneity test
of variances. The K-W ANOVA for winter and summer data of the bins were significant. The
Dunn-Bonferroni test for the *post-hoc* pairwise comparisons shows that the difference of
measured variables among different bins shown in Fig. 10 were significant (Tables S7-S8). The
clear positive relationship between solar radiation and $O_x$ is shown in Fig. 10 (C, D), and the
negative relationship between solar radiation and RH is shown in Fig. 10 (E, F), suggesting
strong atmospheric photochemical activity associated with high $O_x$ periods.

The variations of LO-OOA and MO-OOA showed substantially different patterns with

increases of $O_x$ in winter and summer. In winter, LO-OOA and MO-OOA showed comparable
increasing trends at low $O_x$ level (<35 ppb), with MO-OOA having a stronger response. The
LO-OOA was increased from 0.13 to 0.72 μg m$^{-3}$ when $O_x$ increased from 5 to 35 ppb but
decreased as the $O_x$ increased further. The slope of this increase was approximately 0.023 μg
LO-OOA ppb$^{-1}$ $O_x$. MO-OOA increased from 0.13 to 0.88 μg m$^{-3}$ when the $O_x$ increased from 5
to 35 ppb, with a slope of 0.027 μg MO-OOA ppb$^{-1}$ $O_x$. This leads to a maximum in the mass
fraction of MO-OOA in the mid-$O_x$ level range and also at highest levels of observed $O_x$.

In summer, there is a clear decreasing trend of RH with increases of $O_x$. As discussed



previously, the high level of summertime LO-OOA likely was associated with high LWC.
Therefore, the high mass fraction of LO-OOA at the lowest $O_x$ level (<20 ppb) associated with
the high RH/LWC was likely from aqueous-phase chemistry. After excluding low $O_x$ data (<20
ppb), LO-OOA showed a much stronger response to $O_x$ than did MO-OOA. The summer
LO-OOA showed a significant linear increase from approximately 0.6 to 1.8 μg m$^{-3}$ when $O_x$
increased from 25 to 65 ppb, a slope of 0.03 μg LO-OOA ppb$^{-1}$ $O_x$. This increase was likely in
the case of low RH conditions (<80%, Fig. 8 (D)), when aqueous-phase chemistry did not
promote the formation of LO-OOA (Fig. 8 (H)). Summer MO-OOA increased from 0.36 to
0.67 μg m$^{-3}$ when $O_x$ increased from 25 to 55 ppb but decreased as the $O_x$ increased further.
The slope of this increase was 0.007 μg MO-OOA ppb$^{-1}$ $O_x$. Contrary to winter, LO-OOA
responded more strongly to increases of $O_x$ than MO-OOA did.
The relationship of OOA versus $O_x$ was examined further by excluding nighttime data.
Figure 11 presents the scatter plots of daytime OOA versus $O_x$ for the winter and summer
campaign. The daytime responses of LO-OOA and MO-OOA to $O_x$ in winter were ~1.5 times
that for the whole dataset (Fig. 10 (G, I)), and the increase rate of MO-OOA was higher than
that of LO-OOA. In summer, the slope of the daytime increase of LO-OOA was 1.24 times that
for the whole campaign (Fig. 10 (H)), and the increase rate of daytime MO-OOA was close to
that for whole dataset. These results suggest that the photochemical enhancement of OOA in
winter was more prominent than that in summer. For the summer campaign, the formation of
LO-OOA was more strongly linked to photochemistry compared to MO-OOA. At low
atmospheric oxidative capacity (Ox<20 ppb), aqueous-phase chemistry was likely predominant
in the formation of LO-OOA.





The combined effects of photochemistry and aqueous-phase chemistry on OOA
composition during winter and summer are further demonstrated in Fig. 12. The ratio of
MO-OOA/LO-OOA in winter showed the highest values on the left-top corner in Fig. 12 (A),
suggesting photochemical processing was likely responsible for MO-OOA formation, under
low LWC levels (< 10 µg m$^{-3}$). Additionally, data with high MO-OOA/LO-OOA on the
right-bottom corner in Fig. 12 (A) indicate the important role of aqueous-phase chemistry under
low $O_x$ and high LWC levels. Overall, the concentration of MO-OOA in winter increased as
$O_x$/LWC increased, whereas LO-OOA markedly decreased. This result indicates both
photochemical and aqueous-phase processing played a more important role in enhancing
MO-OOA than LO-OOA in winter. Furthermore, the diurnal patterns of wintertime LO-OOA
only presented a peak value at night while MO-OOA showed one peak value at night (high
LWC) and another one in the afternoon (high $O_x$ period) (Fig. 4).
In summer, data points with low MO-OOA/LO-OOA value on the left-top of Figure 12 (B)
illustrated that LO-OOA was enhanced in high-$O_x$ and low-LWC condition, though the low
MO-OOA/LO-OOA are not confined to just the top left. In case of high LWC level (LWC> 6.5
µg m$^{-3}$), MO-OOA/LO-OOA were much lower (on the right of Figure 12 (B), particularly when
LWC> 10 µg m$^{-3}$). Although MO-OOA increased with LWC and $O_x$, the increase of LO-OOA
was more significant. The effects of both photochemistry (≥ 25 ppb) and aqueous-phase
chemistry (≥ 6.5 µg m$^{-3}$) were more relevant for the formation of LO-OOA than MO-OOA. On
average, the mass concentration of LO-OOA was elevated by nearly 1.2 µg m$^{-3}$ as a ~20 µg
change in LWC (increased from 6.25 µg m$^{-3}$ to 27.5 µg m$^{-3}$, Fig. 8 (H)), which is equivalent to
a 40 ppb change in $O_x$ (increased from 25 ppb to 65 ppb, Fig. 10 (H)). This result further





suggests that the aqueous-phase chemistry is comparable to photochemistry in processing
LO-OOA in summer. The diurnal pattern of summertime LO-OOA displays a peak value at
night and a comparable peak value in the afternoon (Fig. 4).

**4 Conclusions**
Seasonal characterization of NR-PM$_1$ collected using HR-ToF-AMS near Houston in 2014
demonstrated that the mass loading, diurnal patterns, and important formation pathways of
NR-PM$_1$ vary seasonally. The OA was the largest component of NR-PM$_1$ mass, on average,
accounting for 46% and 55% of the mass loadings in winter and summer, respectively, which is
less than that in the north part of Houston, which is influenced by high biogenic emission rates.
Inorganic nitrate was the second largest component in winter (17%) but accounted for only
~0.4% of NR-PM$_1$ mass in summer; SO$_4^{2-}$ was the third and second largest component in
winter (20%) and summer (31%), respectively. ON, on average accounted for ~15 and ~37 %
of OA during winter and summer campaign, respectively. The summertime ON correlated very
well with LO-OOA and concurrently peaked at nighttime. It is likely that ON from NO$_3$ ˙
-initiated oxidation of BVOC in the forested northeastern Houston contributed greatly to
nighttime LO-OOA in summer.
Contributions of factors to wintertime and summertime OA show distinct differences. For
wintertime OA, on average, BBOA, MO-OOA, and COA made similar contributions of 24%,
23% and 22% to total OA mass, respectively. LO-OOA accounted for 18% of OA mass,
followed by HOA. In the summer, LO-OOA represented the largest fraction of the OA mass, 53%
on average. The second largest contributor was MO-OOA (24%). Together, POA constituted



more than half of OA mass (59%) in winter, while it accounted for 23% of OA mass in summer,
highlighting the enhanced impact of primary emissions on OA level during wintertime.
Secondary aerosols account for ~76% and 89% of NR-PM$_1$ mass in winter and summer,
respectively, indicating NR-PM$_1$ mass was likely driven mostly by secondary aerosol
formation.

The two proxies of SOA (LO-OOA and MO-OOA) presented seasonal differences in their

spectral patterns, oxidation degrees and contributions to SOA. MO-OOA showed a higher
contribution to SOA than LO-OOA in winter (56% vs. 44%). In contrast, LO-OOA dominated
SOA in summer (69%). Our results indicate that both photochemical and aqueous-phase
chemistry played important roles in the formation of MO-OOA and LO-OOA. Aqueous-phase
processing likely has strong positive impact on the formation of MO-OOA in the two seasons,
especially in winter. The relationships between MO-OOA and LWC were 0.0076 and 0.0045 μg
MO-OOA μg$^{-1}$ LWC during winter and summer, respectively. Wet removal likely limits
MO-OOA when LWC exceeds 100 μg m$^{-3}$ in winter. Interestingly, the relative importance of
aqueous-phase chemistry versus photochemistry in processing LO-OOA was dependent on RH.
Aqueous-phase processing likely facilitated the local formation of wintertime LO-OOA at low
LWC level (<17.5 μg m$^{-3}$, RH<80%), with a stronger dependence (0.033 μg LO-OOA μg$^{-1}$
LWC) than MO-OOA. In summer, the formation of LO-OOA was enhanced by aqueous-phase
processing at relatively high LWC level (>6.25 μg m$^{-3}$, RH>80%) with a slope of 0.0526 μg
LO-OOA μg$^{-1}$ LWC, while LO-OOA was likely transported non-aqueous regional OOA when
LWC < 6.25 μg m$^{-3}$. These increases of OOA in response to LWC were greatly enhanced
during nighttime. Aqueous-phase chemistry also was predominant in the formation of



summertime LO-OOA at low atmospheric oxidative capacity ($O_x$ < 20 ppb). In general,
summertime LO-OOA showed a much stronger response to $O_x$ than did MO-OOA, with a slope
of 0.0299 µg LO-OOA ppb$^{-1}$ $O_x$. LO-OOA in summer was elevated by nearly 1.2 µg m$^{-3}$ as a
~20 µg change in LWC, which is equivalent to a 40 ppb change in $O_x$.

**Acknowledgments**

The authors would like to acknowledge Yele Sun (Institute of Atmospheric Physics,

Chinese Academy of Sciences) for providing the aq-OOA mass spectra, and Qiao Zhu (Peking
University Shenzhen Graduate School) for assistance in the calculation of organic nitrates and
PMF analysis. The scholarships provided by China Scholarship Council to Qili Dai and
Xiaohui Bi are gratefully acknowledged. Support of the Houston Endowment in development
and deployment of the MAQL also is gratefully acknowledged. Datasets are available by
contacting the corresponding author.

*Author contribution*. Qili Dai performed the data analysis and wrote the manuscript. Robert J.
Griffin and Yinchang Feng assisted heavily with manuscript development and editing. Henry
W. Wallace, Alexander A.T. Bui, James H. Flynn, Barry L. Lefer contributed to data collection
during the field campaigns. Benjamin C. Schulze, Henry W. Wallace, Alexander A.T. Bui and
Nancy P. Sanchez contributed with data analysis. Xiaohui Bi, Benjamin C. Schulze, Alexander
A.T. Bui, Fangzhou Guo, Nancy P. Sanchez, James H. Flynn provided helpful comments and
edits.




*Competing interests.* The authors declare that they have no conflict of interest.

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



**Table 1** Statistics of meteorological parameters, gas-phase pollutants, NR-PM$_1$ species, and PMF OA
factors for the winter and summer campaigns at UHSL.

| Variables | | Season | Ave. value ± 1 SD | Minimum value | Maximum value |
|---|---|---|---|---|---|
| Meteorological parameters | Temp (°C) | Winter | 9.3 ± 6.0 | 0.7 | 25.9 |
| | | Summer | 23.6 ± 3.8 | 12.2 | 33.1 |
| | RH (%) | Winter | 76 ± 18 | 23 | 99 |
| | | Summer | 72 ± 19 | 21 | 98 |
| | WS (m s$^{-1}$) | Winter | 2.1 ± 1.4 | $6.8\times10^{-3}$ | 9.4 |
| | | Summer | 2.1 ± 1.2 | $9.0\times10^{-3}$ | 6.7 |
| | Radiometer (W m$^{-2}$) | Winter | 0.6 ± 0.9 | 0.02 | 3.6 |
| | | Summer | 1.1 ± 1.3 | 0.02 | 4.6 |
| Gas-phase pollutants (ppb) | O$_3$ | Winter | 23.0 ± 12.6 | 0.12 | 53.0 |
| | | Summer | 34.9 ± 15.3 | 0.02 | 75.9 |
| | CO | Winter | 238.7 ± 71.9 | 98.5 | 621.1 |
| | | Summer | 168.3 ± 75.5 | 103.6 | 1110.2 |
| | SO$_2$ | Winter | 1.0 ± 1.9 | $5.7\times10^{-3}$ | 29.5 |
| | | Summer | 0.7 ± 1.7 | $2.8\times10^{-3}$ | 30.9 |
| | NO | Winter | 4.3 ± 6.4 | $2.0\times10^{-3}$ | 74.9 |
| | | Summer | 1.3 ± 4.6 | 0.01 | 68.1 |
| | NO$_2$ | Winter | 12.5 ± 9.7 | 0.8 | 101.2 |
| | | Summer | 4.6 ± 6.4 | 0.2 | 44.4 |
| | NO$_y$ | Winter | 22.9 ± 19.6 | 2.8 | 210.9 |
| | | Summer | 8.6 ± 11.9 | 1.3 | 123.9 |
| NR-PM$_1$ species (µg m$^{-3}$) | OA | Winter | 2.3 ± 1.4 | 0.42 | 9.4 |
| | | Summer | 1.7 ± 1.4 | 0.27 | 12.3 |
| | Sulfate | Winter | 1.4 ± 0.8 | 0.05 | 3.4 |
| | | Summer | 1.3 ± 0.6 | 0.02 | 5.6 |
| | Nitrate | Winter | 1.4 ± 1.4 | 0.02 | 6.9 |
| | | Summer | 0.08 ± 0.1 | 0.01 | 0.9 |
| | Ammonium | Winter | 0.9 ± 0.6 | BDL[a] | 2.8 |
| | | Summer | 0.5 ± 0.2 | 0.02 | 1.8 |
| | Chloride | Winter | 0.06 ± 0.09 | BDL | 1.1 |
| | | Summer | 0.02 ± 0.02 | BDL | 0.5 |
| OA factors (µg m$^{-3}$) | HOA | Winter | 0.4 ± 0.4 | 0[b] | 8.6 |
| | | Summer | 0.3 ± 0.8 | 0 | 10.9 |
| | BBOA | Winter | 0.7 ± 0.7 | 0 | 3.7 |
| | | Summer | 0.2 ± 0.4 | 0 | 5.4 |
| | COA | Winter | 0.7 ± 0.6 | 0 | 4.8 |
| | LO-OOA | Winter | 0.6 ± 0.6 | 0 | 2.1 |
| | | Summer | 1.2 ± 1.3 | 0 | 6.7 |
| | MO-OOA | Winter | 0.7 ± 0.4 | 0 | 1.8 |
| | | Summer | 0.6 ± 0.3 | 0 | 1.6 |

[a]BDL: below detection limit; [b]Statistically determined factor concentrations with values below $1.0\times10^{-3}$ are listed
as 0.



**Table** 2 Correlation (*r*) of OOA mass spectra with previously published spectral database.
(http://cires1.colorado.edu/jimenez-group/HRAMSsd/)

| Factor | Winter | | Summer | | Reference |
|---|---|---|---|---|---|
| | **MO-OOA** | **LO-OOA** | **MO-OOA** | **LO-OOA** | |
| aq-OOA[a] | **0.96** | 0.75 | **0.96** | **0.95** | Sun et al., 2016 |
| MO-OOA | 0.85 | 0.87 | 0.89 | 0.77 | Setyan et al., 2012 |
| MO-OOA | 0.98 | 0.92 | 0.98 | 0.60 | Hu et al., 2015 |
| LV-OOA | 0.97 | 0.91 | 0.98 | 0.62 | Crippa et al., 2013a |
| SV-OOA | 0.65 | 0.70 | 0.70 | 0.78 | Crippa et al., 2013a |
| LO-OOAI, Biogenic-origin | 0.83 | 0.84 | 0.86 | 0.76 | Hu et al., 2015 |
| LO-OOAII, Anthropogenic-origin | 0.78 | 0.80 | 0.82 | 0.74 | Hu et al., 2015 |

[a]aq-OOA is an aqueous-phase-processed SOA reported by Sun et al. (2016); LV=less volatility; SV=semi-volatile.

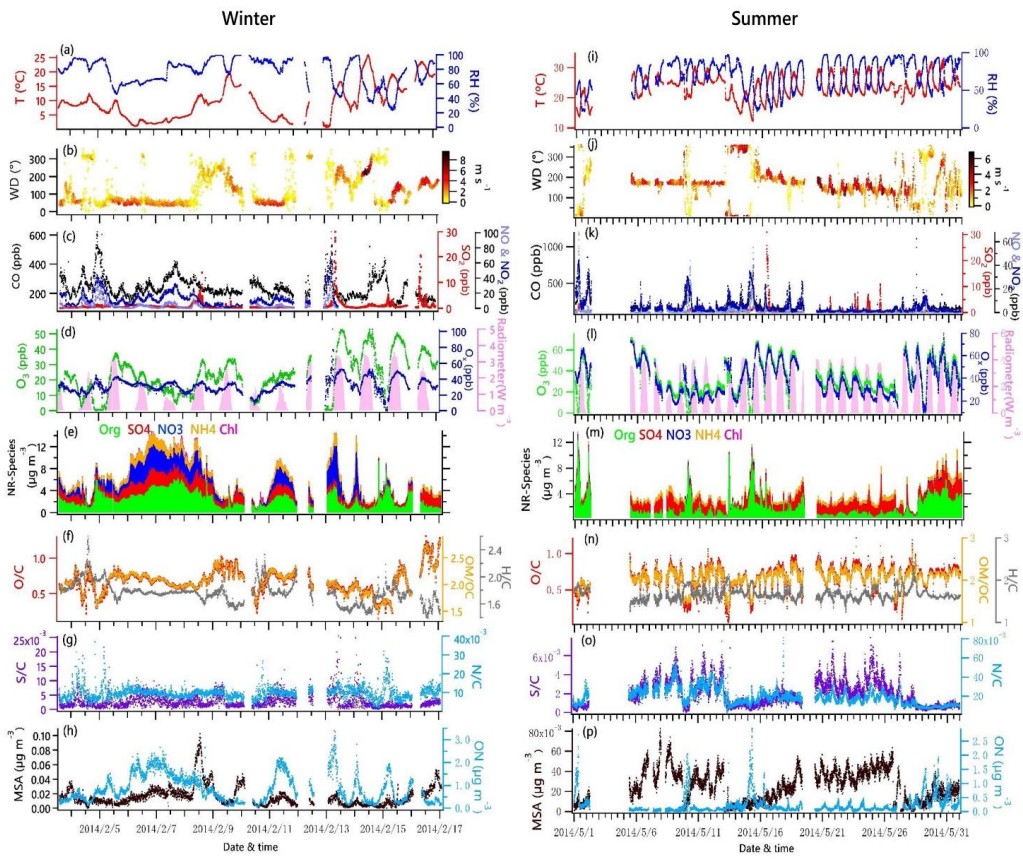


**Figure 1.** Time series of data collected at UHSL in Houston during the sampling periods in
winter and summer 2014. Time series of 5-min average campaign data for **(a, i)** ambient
temperature (T) and relative humidity (RH); **(b, j)** wind direction (WD), with colors showing
different wind speeds (WS); **(c, k)** CO, $SO_2$, NO and $NO_2$; **(d, l)** $O_3$, $O_x$ ($NO_2+O_3$) and solar
radiometer; **(e, m)** NR-$PM_1$ species, including OA, $NO_3^-$, $SO_4^{2-}$, $NH_4^+$, and $Cl^-$; **(f, n)** O/C,
OM/OC, and H/C of OA; **(g, o)** N/C and S/C ratios of OA; and **(h, p)** estimated MSA and ON
concentrations.





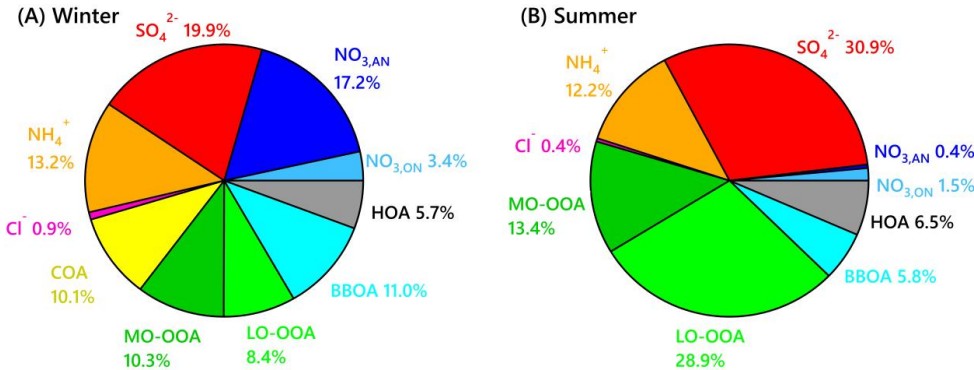


**Figure 2.** Average composition of NR-PM$_1$ species during the winter (**A**) and summer
campaign (**B**) at UHSL.

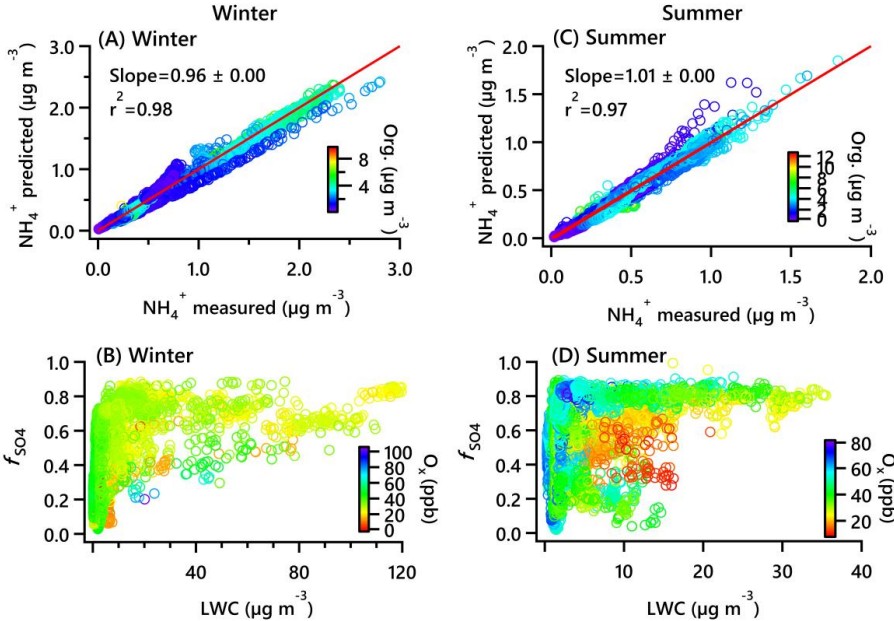


**Figure 3. (A, C)** Scatterplots of predicted NH$_4^+$ vs. measured NH$_4^+$ concentrations over the
winter and summer campaigns. The predicted values were estimated assuming full
neutralization of the HR-ToF-AMS-measured SO$_4^{2-}$, inorganic NO$_3^-$ and Cl$^-$. The data points
were colored by organic concentrations, and the red line is the 1:1 relationship. **(B, D)**
Scatterplots of $f_{SO4}$ vs. LWC, with data points colored by O$_x$ concentrations over the winter and
summer campaigns.





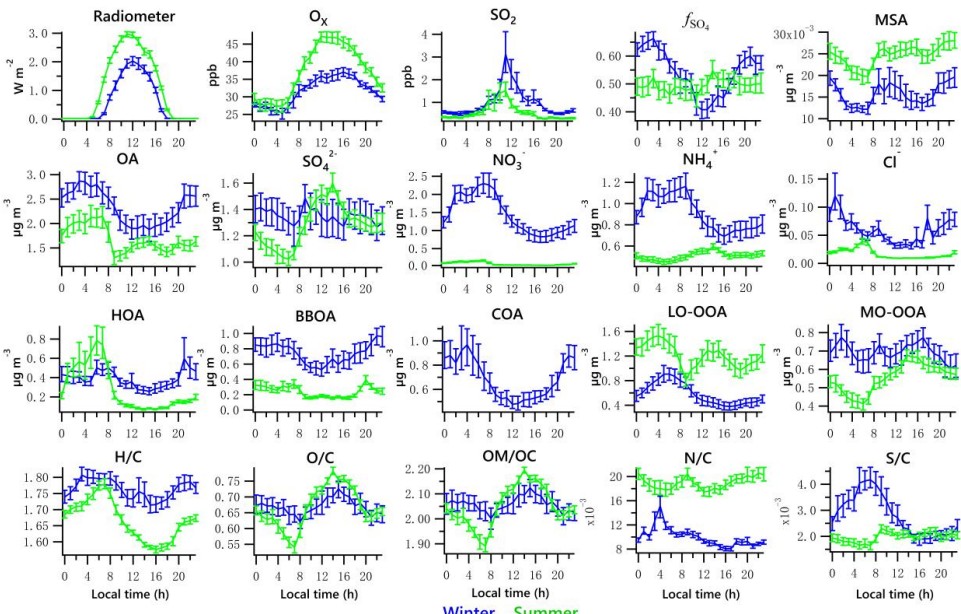


**Figure 4.** Diurnal profiles of radiometer, $O_x$, $SO_2$, $f_{SO4}$, MSA, each of the five NR-PM$_1$ species
(Org, $SO_4^{2-}$, $NO_3^-$, $NH_4^+$ and $Cl^-$), PMF-resolved factors (HOA, BBOA, COA, LO-OOA and
MO-OOA) and elemental ratios (H/C, O/C, OM/OC, N/C and S/C). Lines denote the mean
value, and bars represent the 5/95 percent confidence interval in the mean (blue for winter,
green for summer).




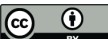

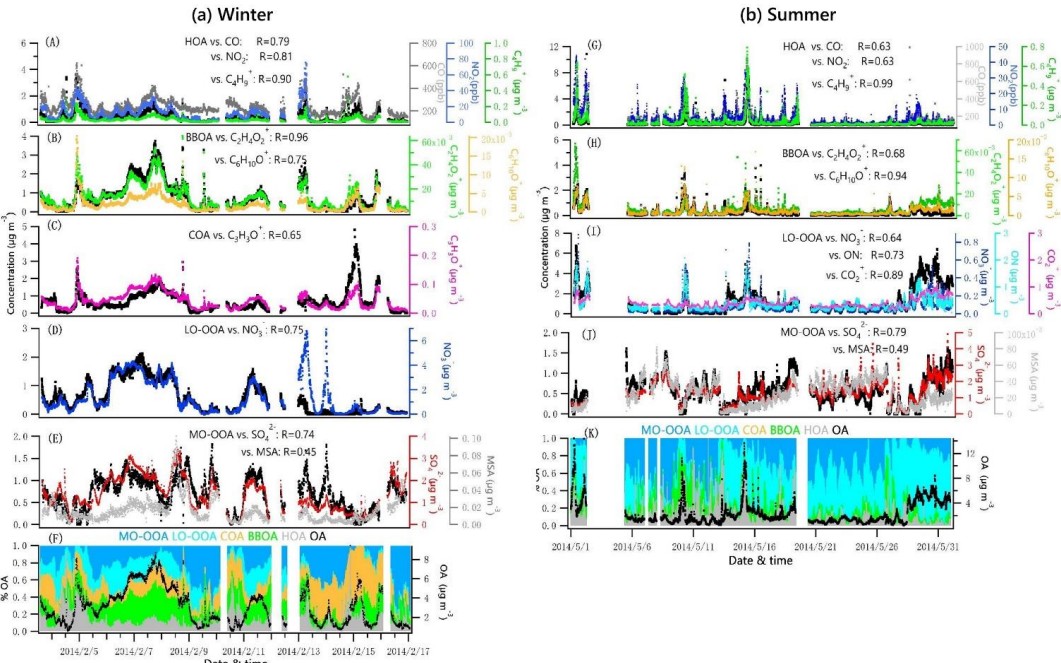


**Figure 5.** Time series of each OA factor and associated correlated species for the winter and
summer campaign at UHSL.





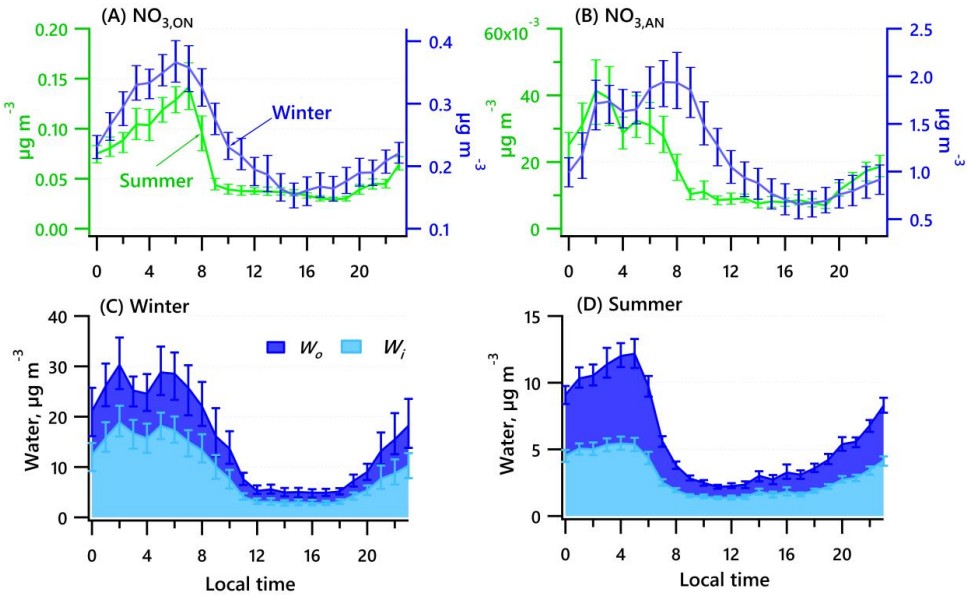


**Figure 6.** Diurnal profiles of nitrate functionality from organic nitrate (**A**) and inorganic nitrate (**B**) for the winter and summer campaigns. Estimated water associated with inorganic and organic aerosol for the winter (**C**) and summer campaigns (**D**). Solid lines denote the mean value (blue for winter, green for summer), and bars represent the 5/95 percent confidence interval in the mean.







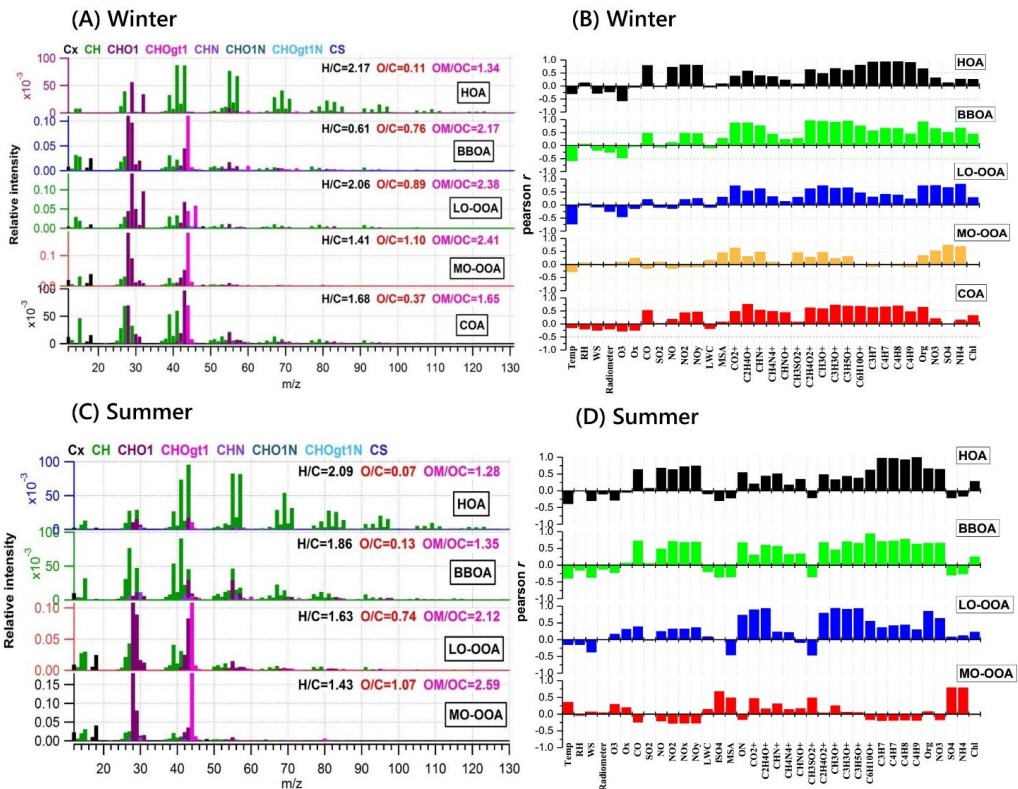


**Figure 7.** Mass spectra of PMF-resolved OA factors (**A**, **C**) and correlation coefficients
between OA factors and other variables (tracer ions, trace gas, meteorological parameters, etc.)
(**B**, **D**) for winter and summer campaigns at UHSL.





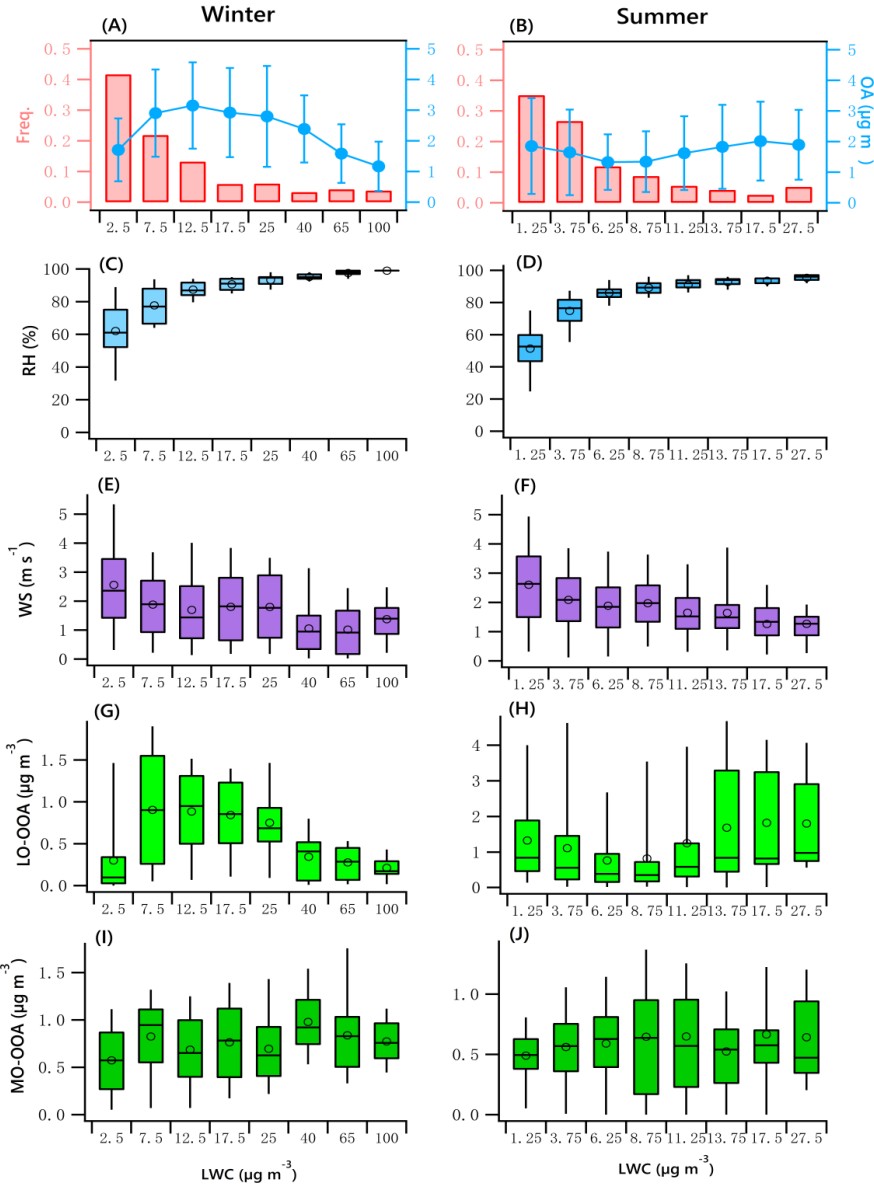

**Figure 8.** OA mass and frequency histograms of data points in each LWC bin for winter (**A**) and summer (**B**). Variations of RH, WS, LO-OOA and MO-OOA mass as a function of LWC in winter (**C, E, G, I**) and summer (**D, F, H, J**). The data were binned according to the LWC (with different increment values), and mean (circle), median (horizontal line), 25[th] and 75[th] percentiles (lower and upper box), and 5[th] and 95[th] percentiles (lower and upper whiskers) are displayed for data in each bin.



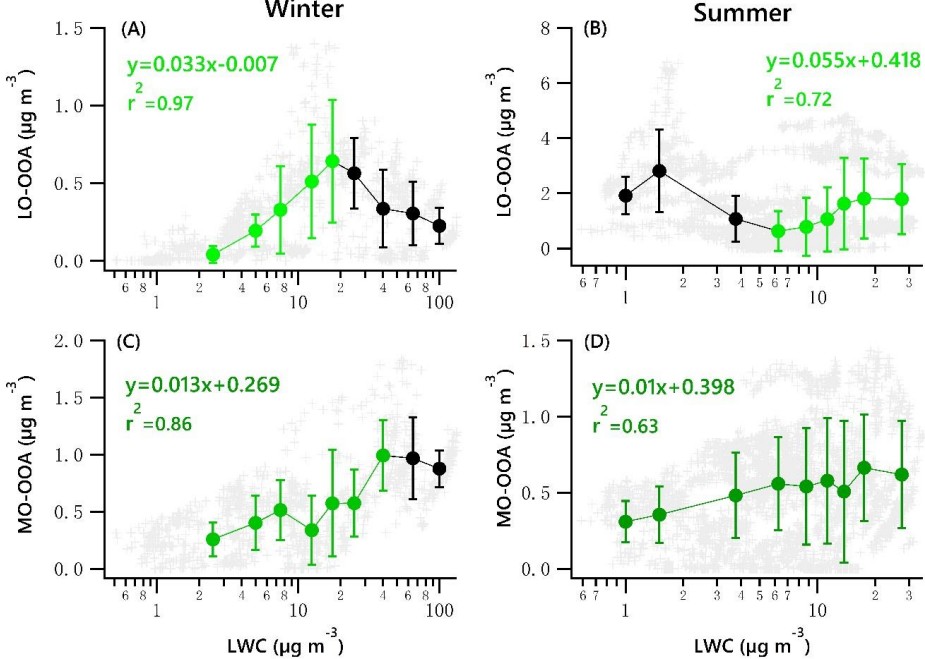

**Figure 9.** Scatter plots of nighttime OOA vs. LWC for the winter and summer campaign. The linear equations are given for fitting only the green dots. Solid dots denote the average value of data in each bin. Bars indicate standard deviations.



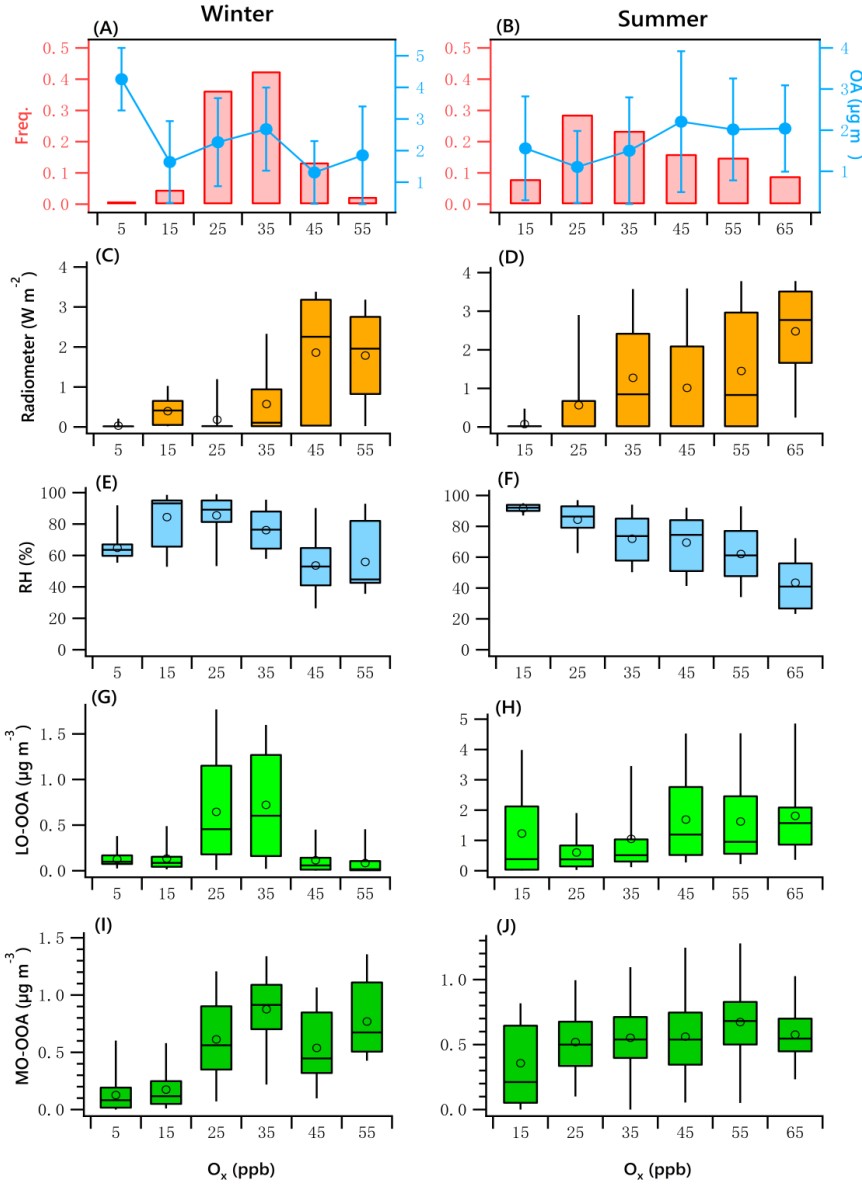

**Figure 10.** OA mass and frequency histograms of data points in each $O_x$ bin for winter (**A**) and summer (**B**). Variations of solar radiation, RH, LO-OOA and MO-OOA mass as a function of LWC in winter (**C, E, G, I**) and summer (**D, F, H, J**). The data were binned according to the $O_x$ (10 ppb increment), and mean (circle), median (horizontal line), 25th and 75th percentiles (lower and upper box), and 5th and 95th percentiles (lower and upper whiskers) are displayed for data in each bin.





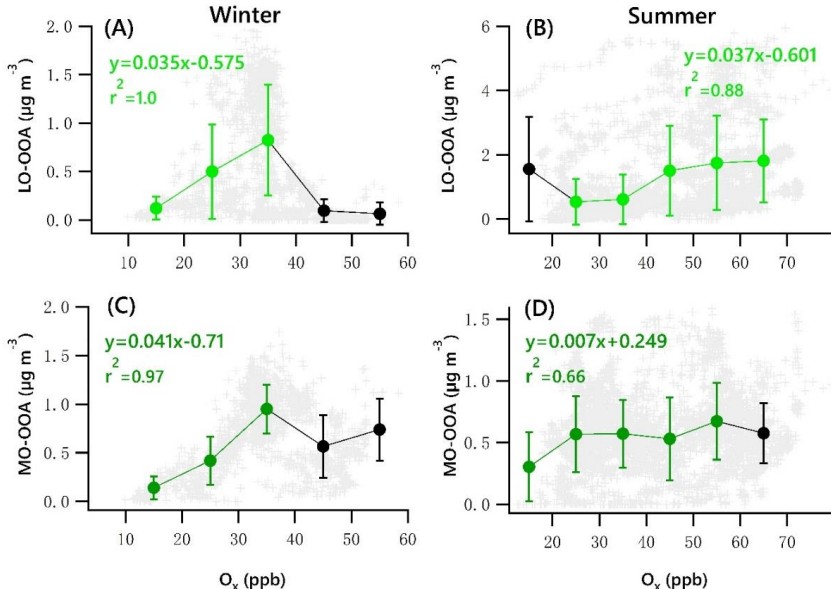

**Figure 11.** Scatter plots of daytime OOA vs. $O_x$ for the winter and summer campaign. The linear equations are given for fitting the green dots. Bars indicate standard deviations.

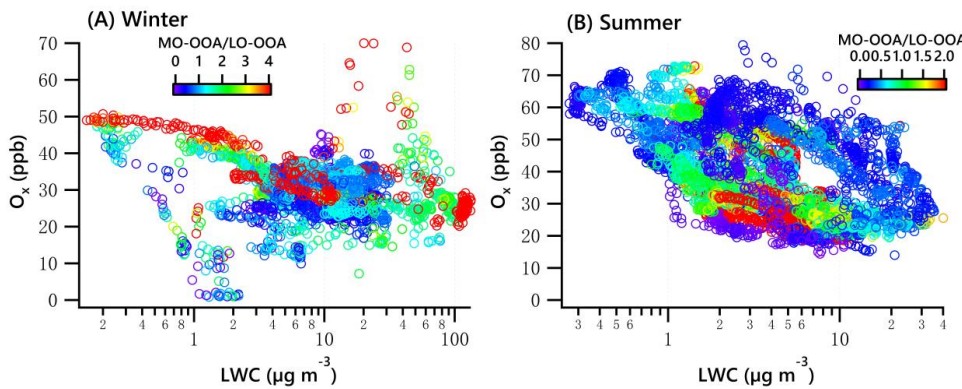

**Figure 12.** $O_x$ vs LWC dependence of the ratio of MO-OOA/LO-OOA in winter (**A**) and summer (**B**).