# Peer review of "Seasonal differences in formation processes of oxidized organic aerosol near"

_Atmospheric Chemistry and Physics, 2018_

## Referee Comment (RC1) · Anonymous Referee #1 · 3 Jan 2019

Review of "Seasonal differences in formation processes of oxidized organic aerosol near Houston, TX" by Q. Dai et al.

**General Comments:**

This manuscript presents measurements of aerosol composition made with an Aerodyne HR-AMS in Houston, TX. The measurements were carried out in two different seasons, and the focus of the analysis is on differences in the OA composition and sources during these different times. Overall, this is a novel data set and the topic is certainly appropriate for *ACP*. The writing is generally good and the manuscript is well-organized. I do have a number of major issues with the manuscript – some addressed here, some in the section below – that prevent me from endorsing it for publication at this time. It may be suitable for publication after a major revision. My greatest concern deals with the analyses and discussion related to Figure 9 (lines 482 - 494) and Figure 11 (lines 550 - 560). This seems like the definition of "cherry picking" data to support one's view, when the entire data set does not. There is no rationale for excluding such large amounts of data until one achieves a good linear fit. It supports the authors' narratives, but I think the conclusions involving these Figures (which are central to the entire manuscript) need substantial revision since they are not consistent with the data.

My additional issues with the manuscript are detailed below:

**Specific Comments:**

- I think that the quantification of the aerosol organic nitrates (ON) have a large uncertainty that needs to be discussed. Equations 3, 4, and 5 indicate the derived ON concentrations are very sensitive to the $R_{ON}$ value. Although the authors have used an $R_{ON}$ value from a very well cited source, there is major uncertainty because the source they cite is based upon a study of SOA from β-pinene oxidation by the nitrate radical. Clearly, the ON formation in this study will be more complex, which adds significant uncertainty to the $R_{ON}$ value and thus to the derived ON concentrations. Much more discussion of this point, including bounds on the ON concentration is warranted.

- The COA factor seems quite problematic given that 1) it is present in winter but absent in summer (cooking is presumably still occurring in the city during this period?), and 2) the diurnal profile of COA (Fig. 4) is inconsistent with both cooking activity and results from many other urban areas.

- The brief discussion on aerosol acidity (Lines 318 - 320) is completely wrong: briefly, the thermodynamic modeling did not include gas-phase ammonia or nitric acid, which significantly limits the ability to characterize acidity. See the extensive body of work from R. Weber and A. Nenes on this topic.

- I had a lot of difficulty with Figure 1 and the associated discussion (lines 349-360). I realize many published papers (including many in *ACP*) follow this standard formula for a paper reporting the results from a ground-based field study. However, I find it almost impossible to actually get anything useful out of Figure 1 – there is simply too much data

presented in too small a space. This is especially true for the discussions about pollutants and wind direction, which cannot be distinguished in Figure 1. If this discussion is central to the manuscript, then additional figures in the Supplemental are likely necessary. If not, then I'd suggest removing (or at least greatly modifying) Figure 1.

- Perhaps this is just a miscalculation, mis-labeled figure or a typo in the manuscript, but the ON concentration estimates (12% and 37% of OA) do not seem consistent with the results in Figure 2, Figure 6, and Table 1. For example, Fig. 2 lists ON contributions to NR-PM1 as 3.4% and 1.5% in winter and summer, respectively. Based on the reported averages (NR-PM1 concentrations of 6 and 3.6 $\mu g/m^3$ in winter and summer, respectively), this would give ON concentrations of 0.204 and 0.054 $\mu g/m^3$. These levels do not seem consistent with Figure 6, nor with the reported contributions to OA.

- All of the discussion about wet removal is misguided (lines 448-452, 468). The authors seem to imply here that the highest levels of aerosol LWC correspond to periods of precipitation. I seriously doubt that is the case, as precipitation events will greatly reduce all of the aerosol species, as well. Either way, the authors should have access to accurate precipitation data, so this point should be backed by evidence rather than speculated upon.

- The discussion linking MSA with aqueous processing is confusing (lines 507 – 520). It is entirely possible for aqueous processing to produce OOA and at the same time for the OOA factors to exhibit weak (or no) correlations with MSA (e.g., if the airmass had a continental origin).

- This is a relatively minor point, but I question the label of "summer" applied to the May measurements. Can the authors use comparison to prior measurement campaigns in Houston to show that May is representative of summertime conditions in terms of source influences, emissions, chemistry, etc.? Further, because of the short duration of the winter measurement period (2 weeks), the limitation that this campaign may not have fully characterized the winter season in Houston should be discussed.

- The paragraph in lines 64-70 seems contradictory with the current results: the reported measurements seem to indicate that Houston is well below the current (and future) standard.

- I understand that it is common to sample an AMS downstream of a Nafion drier (lines 150 - 152), but can the authors comment on potential artifacts from this measurement setup? E.g., the potential loss of semi-volatile organics (see El-Sayed et al., 2016, https://pubs.acs.org/doi/10.1021/acs.est.5b06002).

**Technical Corrections:**

The above issues are substantial enough that any technical corrections can be addressed on review of the revised manuscript.

---

## Referee Comment (RC2) · Anonymous Referee #2 · 6 Jan 2019

General Comments:

The manuscript by Dai et al. characterized seasonal variability of aerosol particles in Houston, Texas using an HR-ToF-AMS. The effects of aqueous-phase and photochemical processing on SOA were investigated. They found that the NR-PM$_1$ mass was driven mostly by secondary aerosol formation regardless of the season. One of the major concerns of this study is the short-period measurements during wintertime which was not statistically significant to explore the effects of aqueous-phase and photochemical processing. Before its publication, the following comments need to be addressed.

Specific Comments:

1. My major concern is that the authors claimed seasonal differences between winter and summer, but the measurements were only made for 2-week period during wintertime with high frequencies of RH>60% (~70% in Fig.1). As a result, there might be a significant uncertainty when comparing the summer and winter data. The authors need to address such uncertainties in the revised manuscript.

2. The uncertainties for the quantification of S/C and N/C
AMS was operated in V-mode (m/∆m=~2000) in this study, separation of N-containing and S-containing are challenging. What are the uncertainties in quantification of S/C and N/C.

3. what the values of CE and RIE used for MSA quantification? Please elaborate.

4. What's the basis of LWC bin division (Fig.8)? Why didn't you use the uniformly-spaced LWC bin as Fig. 10 (Ox binned in 10 ppb).

5. Are there any specific reasons that you used two different versions of software to do data analysis (PIKA1.16 for time series (line 176) and PIKA 1.19D to do PMF analysis (line 235))?

6. How the density of organics was calculated? Was it the I-A method (you mentioned in line 183?) or A-A methods?

7. Line 199-200, NO2+/NO+=0.1-2.0, that is to say, the NO+/NO2+= 0.5-10, which is contradictory with the cited study (5-10) (Xu et al., 2015). What is the ON with R$_{ON}$=0.5? Please mention it here.

8. Line 417, What is the correlation coefficient between LO-OOA/MO-OOA vs. aq-OOA in summer? Are there any further support other than the mass spectra? In fact, the correlation between LO-OOA/MO-OOA vs. SO4 is moderate in Fig. 7.

9. Line 448, the authors attributed the decreased MO-OOA concentration at LWC > 40 µg/m$^3$ to wet removal. How about other species? Besides, in Fig.8 (I), the continuous increase of MO-OOA under LWC < 40 µg/m$^3$ appeared not very clear.

---

## Author Comment (AC1) · 28 Mar 2019

**Response to comments:**

We sincerely thank the reviewers for their helpful comments and guidance. Addressing the major points raised during the review process has substantially improved the quality of the manuscript. In the text that follows, reviewer comments in normal text are followed by author responses in italics.

**Reviewer #1:**

This manuscript presents measurements of aerosol composition made with an Aerodyne HRAMS in Houston, TX. The measurements were carried out in two different seasons, and the focus of the analysis is on differences in the OA composition and sources during these different times. Overall, this is a novel data set and the topic is certainly appropriate for ACP. The writing is generally good and the manuscript is well-organized. I do have a number of major issues with the manuscript – some addressed here, some in the section below – that prevent me from endorsing it for publication at this time. It may be suitable for publication after a major revision.

My greatest concern deals with the analyses and discussion related to Figure 9 (lines 482 - 494) and Figure 11 (lines 550 - 560). This seems like the definition of "cherry picking" data to support one's view, when the entire data set does not. There is no rationale for excluding such large amounts of data until one achieves a good linear fit. It supports the authors' narratives, but I think the conclusions involving these Figures (which are central to the entire manuscript) need substantial revision since they are not consistent with the data.

*Response:*

*We thank the reviewer for highlighting this point, but we respectfully disagree with the reviewer that the data analysis involves cherry picking. Please note the fact that it is possible to find a distinct regression relationship between two observed variables for part of a given data set, in contrast to the whole data set. We can find many examples in the literature: for example, the third figure in Li et al. (2016) and the eighth figure in Guo et al. (2016). In this*

*work, OOA presented a non-linear relationship with LWC for our whole dataset, which is in agreement with a previous study (Xu et al., 2017). The main focus of their work was to investigate the effects of aqueous-phase and photochemical processing on secondary organic aerosol formation and evolution. Their data suggested that the OOA exhibited a non-linear relationship with RH for their whole dataset in different seasons. However, based on least squares regression correlation analysis, Sullivan et al. (2016) reported a relationship of an increasing water-soluble organic carbon with LWC for RH increased from 40% to 70% during a specific short period. Thus, our aim is to examine if there is a linear relationship between OOA and LWC during specific loadings of LWC.*

*Classical regression methods, such as linear least squares regression, were developed to specify a global function to fit a model to an entire data set. However, as mentioned by the reviewer, no rationale for determining what fraction of the data is appropriate for a good linear fit was provided. In general, data are usually chosen arbitrarily. Here we provide the detailed strategy for choosing the appropriate fraction of data points utilized in subsequent local linear regression analyses.*

*The first step is to examine the linear or non-linear relationships between two variables by fitting the given data with a locally weighted scatter plot smoothing algorithm (LOWESS). Further details on the LOWESS technique is available in Cleveland (1981). We take the relationship between LO-OOA and LWC during the summer campaign as an example. The LOWESS function has a "span" argument (f) that represents the proportion of the total number of points that contribute to each local fitted value. The fitting curve is constructed by connecting the "fitted" value with lines for each data point, with colors from red to blue to show the effect of the smoothing parameter choice. The red curve (f=0.01, where f is equal to the proportion of points) is "looser" than the blue curve (f=1), as the blue curve is fitted to all data points. As shown in Figure S14(C) (named as in the current SI), the blue curve fitted for data points with LWC concentration great than 5 µg m⁻³ is likely a straight line. To further verify this assumption, we resampled the original data by using a bootstrap method and recalculated the LOWESS curves. Figure S14(D) presents the 400 LOWESS curves for all resampled data. A strong linear relationship between LO-OOA and LWC can be found for data with LWC great than 6 µg m⁻³. Because the number of the corresponding data points*

*decreased as the LWC increased, the fitted curve is sparsely distributed for high LWC data. Although these resampled data points are not as representative compared to the data at lower LWC, the linear relationship is apparent.*

[Figure]

**Figure S14**. LOWESS curves for the nighttime LO-OOA vs. LWC during winter (A) and summer (C) and for the associated resampled data obtained by bootstrap method (B for winter and D for summer).

*Author's changes in manuscript:*

*Our revision to this comment is included in the following bulleted list:*

*1. Added the results of LOWESS analysis into supplementary materials (Figures S14-17).*

*2. Added a paragraph before the discussion of Figure 8 to clarify the potential relationship between OOA and LWC during nighttime:*

*"The potential linear relationship between OOA and LWC for the nighttime data was investigated by fitting the data with a locally weighted scatter plot smoothing algorithm (LOWESS, (Cleveland, 1981)). According to the LOWESS curves fitted for the original nighttime data and the resampled data obtained by a bootstrap method (Figs. S14-15), there likely exists a linear relationship between LO-OOA and LWC for data points with LWC less than 20 $\mu g\ m^{-3}$ and greater than 6 $\mu g\ m^{-3}$ for the winter and summer periods, respectively. As*

*for MO-OOA, such a linear relationship likely exists when LWC is less than 50 and 7 μg m$^{-3}$ for the winter and summer periods, respectively."*

*3. Added another paragraph before the discussion of Figure 10 to clarify the potential relationship between OOA and O$_x$ during daytime:*

*"According to the LOWESS curves fitted for the original daytime data and resampled data obtained using a bootstrap method (Figs. S16-17), there likely exists a linear relationship between LO-OOA and O$_x$ when O$_x$ is less than 35 ppb and greater than 20 ppb for the winter and summer periods, respectively. As for MO-OOA, the linear relationship likely exists for data points with O$_x$ less than 35 ppb for the winter period, but the linear relationship is less prominent."*

*4. Deleted the discussion of the linear relationship between MO-OOA and O$_x$ for the summer campaign.*

**Specific comment 1:**

I think that the quantification of the aerosol organic nitrates (ON) have a large uncertainty that needs to be discussed. Equations 3, 4, and 5 indicate the derived ON concentrations are very sensitive to the $R_{ON}$ value. Although the authors have used an $R_{ON}$ value from a very well cited source, there is major uncertainty because the source they cite is based upon a study of SOA from β-pinene oxidation by the nitrate radical. Clearly, the ON formation in this study will be more complex, which adds significant uncertainty to the $R_{ON}$ value and thus to the derived ON concentrations. Much more discussion of this point, including bounds on the ON concentration is warranted.

*Response:*

*Previous studies found that isoprene was the main biogenic VOC in Houston (Leuchner and Rappengluck, 2010; Kota et al., 2014), and Brown et al. (2013) reported that monoterpenes and isoprene were frequently present within the nocturnal boundary layer in the Houston area and underwent rapid oxidation, mainly by nitrate radical. Given the large abundance of monoterpene and isoprene in the Houston area, similar to Xu et al. (2015), we assume organic nitrates formed via isoprene and beta-pinene oxidation are representative. Fry et al.*

*(2013) assumed that the $R_{ON}/R_{NH4NO3}$ value is instrument-independent, and further estimated the average $R_{ON}/R_{NH4NO3}$ of 2.25 for the organic nitrate standards. The $R_{ON}/R_{NH4NO3}$ values vary with precursor VOC. We utilized the average $R_{ON}/R_{NH4NO3}$ of isoprene (2.08, (Bruns et al., 2010)) and beta-pinene organic nitrates (3.99, (Boyd et al., 2015)) from the literature to obtain an estimate range of $R_{ON}$ by using the $NO_x^+$ method. The mass range of ON is estimated by assuming that the average molecular weights of organic molecules with nitrate functional groups are 200 to 300 g $mol^{-1}$ (Surratt et al., 2008; Rollins et al., 2012).*

*The result of estimated ON is available in Table S2. The associated Figures and content in the original manuscript were updated accordingly. Here we retain the results estimated with $R_{ON}$ value of 0.166 in the manuscript.*

**Table S2**. Results of organic nitrates estimated using the $NO_x^+$ ratio method.

| | $NO_{3,ON}$ conc. ($\mu g\ m^{-3}$) | | $NO_{3,ON}/NO_{3,obs}$ | | ON/OA | |
|---|---|---|---|---|---|---|
| | lower | upper | lower | upper | lower | upper |
| Winter | 0.22 | 0.34 | 34% | 35% | 31% | 66% |
| Summer | 0.05 | 0.06 | 61% | 81% | 9% | 17% |

**Specific comment 2:**

The COA factor seems quite problematic given that 1) it is present in winter but absent in summer (cooking is presumably still occurring in the city during this period?), and 2) the diurnal profile of COA (Fig. 4) is inconsistent with both cooking activity and results from many urban areas.

*Response:*

*1) There is a restaurant situated directly northeast of the measurement site (UHSL). The northeasterly winds were observed at the measurement site with a high frequency during the winter campaign but not the summer (Fig. 1), which is likely to be responsible for the impact of emissions from cooking activities on this site during winter. It also is possible that increased processing in the summer led COA to be oxidized and included in one of the OOA factors.*

*2) In our previous supplemental material, we have interpreted the COA factor by comparing the factor mass spectra with other factors, examining the relationship of COA factor versus cooking-tracer ion ($C_3H_3O^+$) and investigating the factor's signal ratio of m/z 55 to m/z 57. As shown in the following figure (now S13), the signals for m/z 55 to m/z 57 for COA in the summer is close to that for LO-OOA and MO-OOA, and higher than that for BBOA and HOA. This is a strong evidence for the interpretation of the COA factor (Mohr et al., 2012).*

[Figure]

**Figure S13.** $f_{55}$ vs. $f_{57}$ of PMF factors for the winter and summer periods. (w) and (s) denote the winter and summer data, respectively.

*Although there is no routine peak during mealtime in the diurnal pattern of COA, the COA factor mass spectra correlated moderately with previously reported COA factors deduced from PMF analysis (as shown in the table below), which further supports the interpretation of COA factor in this work.*

**Table** Correlation ($r$) of COA mass spectra with previously published spectral database. (http://cires1.colorado.edu/jimenez-group/HRAMSsd/)

| References | Study area | $r$ |
|---|---|---|
| (Mohr et al., 2012) | Barcelona | 0.76 |
| (Crippa et al., 2013) | Paris | 0.58 |

| (Elser et al., 2016) | Xi'an and Beijing | 0.65 |
| (Hu et al., 2016) | Beijing | 0.65 |

**Specific comment 3:**

The brief discussion on aerosol acidity (Lines 318-320) is completely wrong: briefly, the thermodynamic modeling did not include gas-phase ammonia or nitric acid, which significantly limits the ability to characterize acidity. See the extensive body of work form R. Weber and A. Nenes on this topic.

*Response:*

*The original sentence (original lines 318-320) has been deleted.*

**Specific comment 4:**

I had a lot of difficulty with Figure 1 and the associated discussion (Lines 349-360). I realize many published papers (including many in ACP) follow this standard formula for a paper reporting the results from a ground-based field study. However, I find it almost impossible to actually get anything useful out of Figure 1-there is simply too much data presented in too small a space. This is especially true for the discussions about pollutants and wind direction, which cannot be distinguished in Figure 1. If the discussion is central to the manuscript, then additional figures in the Supplemental and likely necessary. If not, then I'd suggest removing (or greatly modifying) Figure 1.

*Response:*

*For the sake of clarity, we have removed the panels of elemental ratios (O/C, H/C, OM/OC, and N/C), ON and MSA from the stacked time series plot and put these panels into the supplemental material (Figure S2).*

**Specific comment 5:**

Perhaps this is just a miscalculation, mis-labeled figure or a typo in the manuscript, but the ON concentration estimates (12% and 37% of OA) do not seem consistent with the results in Figure 2, Figure 6, and Table 1. For example, Fig. 2 lists ON contributions to NR-PM$_1$ as 3.4% and 1.5% in winter and summer, respectively. Based on the reported averages (NR-PM$_1$ concentrations of 6 and 3.6 µg/m$^3$ in winter and summer, respectively), this would give ON concentrations of 0.204 and 0.054 µg/m$^3$. These levels do not seen consistent with Figure 6, nor with the reported contributions to OA.

*Response:*

*We apologize for the error. As suggested, we estimated the bounds of ON based on the $R_{ON}$ values (and their precursor VOCs) that are relevant to the Houston area in the revised manuscript. The estimated result is available in Table S2. The ON concentrations in Tables, Figures and text have been updated in the revised manuscript.*

**Specific comment 6:**

All of the discussion about wet removal is misguided (lines 448-452, 468). The authors seem to imply here that the highest levels of aerosol LWC correspond to periods of precipitation. I seriously doubt that is the case, as precipitation events will greatly reduce all of the aerosol species, as well. Either way, the authors should have access to accurate precipitation data, so this point should be backed by evidence rather than speculated upon.

*Response:*

*Precipitation totals from a nearby Texas Commission on Environmental Quality (TCEQ) monitor site were added to Figure 1 (b, g). The periods of precipitation events correspond to high levels of RH. Indeed, the wet removal effect works on all species. We have rephrased the statement on wet removal effect in response to this comment.*

*Lines 436-441: "This result indicates that wet removal may dominate under an extremely high RH environment coupled with stagnant air (WS < 2 m/s, Fig. 7(E)), as the OA concentration decreased at extremely high LWC level (Fig. 7(A)). In summer, the OA mass decreased when LWC increased from 1.25 to 6.25 µg m$^{-3}$ but increased when LWC increased*

*further, suggesting the wet removal effect is likely not as strong as that in winter because of the relatively lower LWC in summer."*

**Specific comment 7:**

This discussion linking MSA with aqueous processing is confusing (lines 507-520). It is entirely possible for aqueous processing to produce OOA and at the same time for the OOA factors to exhibit weak (or no) correlations with MSA (e.g., if the air mass had a continental origin).

*Response:*

*The discussion linking MSA with aqueous processing supports the idea that the summertime MO-OOA formation was more likely associated with aqueous processing than LO-OOA, as MO-OOA positively correlated with MSA but LO-OOA exhibited weak correlation with MSA. This result also indicates that MO-OOA was impacted by marine aerosol and that LO-OOA likely originated from continental areas.*

**Specific comment 8:**

This is a relatively minor point, but I question the label of "summer" applied to the May measurements. Can the authors use comparison to priori measurement campaigns in Houston to show that May is representative of summertime conditions in terms of source influences, emissions, chemistry, etc.? Further, because of the short duration of the winter measurement period (2 weeks), the limitation that this campaign may not have fully characterized the winter season in Houston should be discussed.

*Response:*

*We agree with the reviewer that the sampling periods are too short to cover the whole seasons thus we have added a comment in Section 2.1:*

*"The data collected during winter campaign are limited in duration; thus, the following discussion focuses primarily on the summer campaign. The label of "winter/summer" in the text denotes the measurement period in the winter/summer."*

*The fourth paragraph of the Introduction section has been deleted, and we have shortened the discussion about winter data in the text. It should also be noted that even though May officially is part of spring, the temperatures in Texas in May are high enough that they are much more characteristic of summer meteorology.*

**Specific comment 9:**

The paragraph in lines 64-70 seems contradictory with the current results: the reported measurement seem to indicate that Houston is well below the current (and future) standard.

*Response:*

*The original statement has been deleted.*

**Specific comment 10:**

I understand that it is common to sample an AMS downstream of a Nafion drier (lines 150-152), but can the authors comment on potential artifacts from this measurement setup? E.g., the potential loss of semi-volatile organics.

*Response:*

*The work of El-Sayed et al. (2016) indicates that drying of aerosol water led to the evaporation of condensed-phase organics (for both daytime and nighttime sampling periods). For the purposes of this study, therefore, we may be underestimating the contribution of aqueous-SOA (i.e., IEPOX as an example in El-Sayed et al. (2016)).*
*We have added comment in Section 2.3.2:*
*"As suggested by El-Sayed et al. (2016), drying of aerosol water may have led to the evaporation of condensed-phase organics. Thus, the mass concentrations of resolved OA factors here are a lower-bound, conservative estimate due to potential losses of aqueous-SOA in the Nafion dryer element."*


[revised manuscript text omitted]

**Reviewer #2:**

**Specific Comment 1:**

My major concern is that the authors claimed seasonal differences between winter and summer, but the measurements were only made for 2-week period during wintertime with high frequencies of RH>60% (~70% in Fig.1). As a result, there might be a significant uncertainty when comparing the summer and winter data. The authors need to address such uncertainties in the revised manuscript.

*Response:*

*Please see response to similar comment from Reviewer #1.*

**Specific Comment 2:**

The uncertainties for the quantification of S/C and N/C AMS was operated in V-mode (m/Δm=~2000) in this study, separation of N-containing and S-containing are challenging. What are the uncertainties in quantification of S/C and N/C.

*Response:*

*Because of the lack of measurement results for standard organic nitrates and organic sulfates, we are unable to estimate quantitatively the uncertainties in N/C and S/C. Since the elemental ratio packages in PIKA indicate that the calibration factors for S/C ratios are "not measured/published", we have deleted the S/C data in the revised manuscript.*

*The uncertainties in quantification of N/C include: 1) the important nitrogen-containing ion ($CH_2N^+$, m/z 28 (Ge et al., 2017)) was excluded from the AMS elemental analysis due to the overwhelming interference of adjacent $N_2^+$ ion, resulting in the current N/C ratio being underestimated by ~20% on average (Struckmeier et al., 2016); 2) because the signals of $C_xH_yN_p^+$ and $C_xH_yO_zN_p+$ are much lower than $C_xH_y^+$ and $C_xH_yO_z^+$, the determination of N/C relies on the mass resolution (m/Δm). The mass resolution of V-mode AMS is just half of that of W-mode, making separation and quantification of the nitrogen-containing ions above m/z 50 impossible (Xu et al., 2017); and 3) N-containing ion peaks are very often on tails of larger peaks, thus small errors in m/z calibration can generate uncertainty in estimated N/C. Aiken et al. (2007) compared the N/C ratios for methylamine, ethylamine, and hydrogen cyanide from a NIST electron-ionization database and from elemental analysis using an AMS. The analysis of AMS (NIST) spectra indicates that quantification of N/C is possible with an average error of 20%.*

**Specific Comment 3:**

What the values of CE and RIE used for MSA quantification? Please elaborate.

*Response:*

*Following previous studies by Zorn et al. (2008) and Huang et al. (2015), the relative ionization efficiency (RIE) of MSA (1.3) was assumed to be the average of the value for organic species (RIEorg = 1.4) and sulfate species (RIEso4 = 1.2). The collection efficiency of all ions composing MSA was calculated using the composition-dependent collection efficiency developed by Middlebrook et al. (2015). The resulting collection efficiency was 0.5 for 7.3% and 4.2% of time of the summer and winter campaign, respectively, and was 1.0 for the remaining time of the two campaigns.*

**Specific Comment 4:**

What's the basis of LWC bin division (Fig.8)? Why didn't you use the uniformly-spaced LWC bin as Fig. 10 (Ox binned in 10 ppb).

*Response:*

*Unlike the $O_x$ concentration, the LWC in the winter and summer show a power function distribution and a lognormal distribution, respectively. We used uniformly-spaced LWC bins for most of the data. The summer data are binned in 5 $\mu g$ $m^{-3}$ increments from 0 to 20 $\mu g$ $m^{-3}$ (covering 83% of data points), and winter data are binned in 2.5 $\mu g$ $m^{-3}$ increments from 0 to 15 $\mu g$ $m^{-3}$ (covering 91% of data points). The number of data points within each bin is too small to produce a reasonable result if the remaining data at higher values are binned using the same increment. This data processing method is common in literature (Huang et al., 2015).*

[Figure]

**Figure** The frequency histograms of LWC in the winter and summer.

**Specific Comment 5:**

Are there any specific reasons that you used two different versions of software to do data analysis (PIKA1.16 for time series (line 176) and PIKA 1.19D to do PMF analysis (line 235))?

*Response:*

*We performed updated data analysis in the version of PIKA 1.19D, and all data presented in Figures, Tables and the text have been updated in the revised manuscript.*

**Specific Comment 6:**

How the density of organics was calculated? Was it the I-A method (you mentioned in line 183?) or A-A methods?

*Response:*

*The organic density was estimated using an empirical equation based on elemental ratios (Kuwata et al., 2012):*

$$\rho_{Org} = 1000 \times \left[ \frac{12 + \frac{H}{C} + 16 \times \frac{O}{C}}{7.0 + 5 \times \frac{H}{C} + 4.15 \times \frac{O}{C}} \right]$$

*The elemental ratios were estimated using the updated I-A method (Canagaratna et al., 2015).*

**Specific Comment 7:**

Line 199-200, $NO_2^+/NO^+ = 0.1-2.0$, that is to say, the $NO^+/NO_2^+ = 0.5-10$, which is contradictory with the cited study (5-10) (Xu et al., 2015). What is the ON with $R_{ON}=0.5$? Please mention it here.

*Response:*

*This is an error. The $NO_2^+/NO^+$ ratio is in the range of 0.1-0.2. This mistyping has been deleted and the bounds of ON have been estimated. Please see response to Reviewer #1 regarding uncertainties in ON.*

**Specific Comment 8:**

Line 417, what is the correlation coefficient between LO-OOA/MO-OOA vs. aq-OOA in summer? Are there any further support other than the mass spectra? In fact, the correlation between LO-OOA/MO-OOA vs. $SO_4$ is moderate in Fig. 7.

*Response:*

*The correlation coefficients of summertime LO-OOA/MO-OOA mass spectra with aq-OOA is 0.95/0.96, respectively. This result suggests that the formation of LO-OOA in summer and MO-OOA in both seasons likely involved aqueous processing. The correlation between OOA and MSA also supports this. MSA has a relatively higher correlation coefficient with MO-OOA (r=0.45) compared to LO-OOA (r=0.30), though the correlation also is influenced by many other factors. Additionally, the high correlation coefficients of $SO_4$ with MO-OOA in both seasons (r=0.74 and 0.79 for winter and summer, respectively) further support the conclusion, although the correlation of LO-OOA and $SO_4$ is not as strong as expected. We believe that the strong correlation between the mass spectra of OOA and aq-OOA is strong enough to support our conclusion.*

**Specific Comment 9:**

Line 448, the authors attributed the decreased MO-OOA concentration at LWC > 40 $\mu g/m^3$ to wet removal. How about other species? Besides, in Fig.8 (I), the continuous increase of

MO-OOA under LWC < 40 μg/m$^3$ appeared not very clear.

Response:

*Please see response to Reviewer #1. The continuous increase of MO-OOA under LWC< 40 μg m$^{-3}$ is not very clear, but the maximum and minimum values of each bin increased obviously. We further investigated the relationship between MO-OOA and LWC for nighttime data only, and the MO-OOA shows a significant increase trend under LWC< 40 μg m$^{-3}$ during nighttime.*

---

## Referee Report (RR1)

The authors have continued to improve the manuscript using additional evidence and more appropriate discussions. I recommend publication after one further consideration.

Are there any specific reasons that you used updated I-A method rather than A-A method mentioned by Kuwata et al. (2011) ? Can the authors use comparisons to priori density measurements to show that using I-A method is more reasonable than A-A methods?

References

Kuwata, M., Zorn, S. R., and Martin, S. T.: Using Elemental Ratios to Predict the Density of Organic Material Composed of Carbon, Hydrogen, and Oxygen, Environ. Sci. Technol., 46, 787-794, 10.1021/es202525q, 2011.

---

## Author Response (AR2)

**Response to comments:**

We sincerely thank the reviewer for the helpful comments. In the text that follows, reviewer comments in normal text are followed by author responses in italics.

**Comment 1:**

I urge the authors to reconsider the discussion regarding wet removal (Referee 1 specific comment 6; lines 977-986, 999-1006 in the revised track changes version). In particular, the authors need to more convincingly show that there is a connection between wet removal (rain, fog, etc.) and high LWC (the addition of precipitation to figure 1 does not adequately show this).

Regarding the variation in MO-OOA with LWC (Figure 7), given the large variability in

MO-OOA in each bin, it does not appear to me that there are any robust trends. I urge the authors to consider how robust the trends are and add discussion regarding this to lines 999-1006. Also, could the results be explained by changes in production rather than changes in removal?

*Response:*

*We thank the reviewer for bring this point to our attention again. The total cloud cover (%) from*

*Air Resources Laboratory's data archive on the website of NOAA's READY Archived*

*Meteorology (https://ready.arl.noaa.gov/READYamet.php) for the measurement location were*

*added to Figure 1 (A, F). The relative humidity was likely high when the total cloud cover was*

*significantly high, which could decrease the production of OOA or enhance its removal. Text has*

*been modified appropriately.*

*We have added the correlation coefficient of MO-OOA vs. LWC to denote the robustness of the*

*variation trend in MO-OOA with LWC.*

*Our revisions are included in the following bulleted list:*

*1. The previous discussion (original lines 999-1006) has been revised as follows (lines 450-456*

*in the revised manuscript):*

*"MO-OOA slightly increased during both seasons as LWC increased (Fig. 7 (I, J)). In winter,*

*MO-OOA presents an increasing trend from 0.57 to 0.98 µg m$^{-3}$ when LWC increased from 2.5 to*

*40 µg m$^{-3}$ but decreased as the LWC increased further. The slope of this increase was*

*approximately 0.008 µg MO-OOA µg$^{-1}$ LWC with correlation coefficient of 0.55. In summer,*

*MO-OOA appears to increase from 0.49 to 0.64 µg m$^{-3}$ when LWC increased from 2.5 to 27.5 µg*

*m$^{-3}$, with slope of 0.005 µg MO-OOA µg$^{-1}$ LWC (R$^2$=0.34). In winter, because of the decrease in*

*LO-OOA with LWC, the relative fraction of MO-OOA increases as LWC increases."*

*2. We added another statement regarding the comparison of the varying trend of MO-OOA with*

*LWC during nighttime and whole periods (lines 484-485):*

*"The nighttime increasing trends of MO-OOA against LWC in both seasons are stronger than*

*those shown in Fig. 7(I, J) in terms of the correlation coefficient values."*

*3. We altered phrasing regarding changes in production versus removal in several locations.*

**Comment 2:**

In regards to Referee 1 specific comment 7, I also find the discussion regarding MSA confusing.

It would help if the authors incorporated some of their response to the referee into the revised version of the manuscript.

*Response:*

*Our discussion regarding MSA suggested that the summertime MO-OOA formation was more*

*likely associated with aqueous processing than LO-OOA as MO-OOA positively correlated with*

*MSA but LO-OOA exhibited weak correlation with MSA. However, as suggested by Referee 1*

*specific comment 7, it is entirely possible for aqueous processing to produce OOA and at the*

*same time for the OOA factors to exhibit weak (or no) correlations with MSA (e.g., if the air*

*mass had a continental origin). Thus, we deleted the discussion relevant to MSA in the updated*

*manuscript.*

**Comment 3:**

Lines 894-896: What exactly is meant by "more prominent"? More mass or a larger fraction of total sulfate production? Have the authors considered other explanations that could account for this?

*Response:*

*We have rephrased the sentence as follows on lines 347-350:*

*"By comparing the diurnal plots of sulfate in winter and summer, it appears that sulfate*

*generated from aqueous chemistry accounted for more mass and a greater fraction of total*

*sulfate production in winter than in summer."*

**Comment 4:**

Line 964: "likely results" It should be possible to directly calculate this and make a more
definitive statement.

*Response on lines 412-416:*

*The correlation coefficients have been calculated and added in the statement that provide a more*
*explicit explanation about the statement:*

*"It should be noted that a fit for the binned data likely results in an increase in $R^2$ compared to*
*the fit for the original data. For example, the correlation coefficient of the fit for the averaged*
*binned wintertime MO-OOA (increased from 0.57 to 0.98 $\mu g\ m^{-3}$) versus LWC (increased from*
*2.5 to 40 $\mu g\ m^{-3}$) is 0.55, while it is 0.06 for the original data (Figure 7(I))."*

**Comment 5:**

Several sections of the manuscript need to be revised to clarify that the results are a correlation
and not definitive proof that a specific process is happening. Specific examples include (line
numbers refer to track changes version):

Lines 977-986 (see above comment regarding production vs loss)

*Response:*

*Changes have been made throughout the document to achieve these clarifications and soften the*
*language used.*

**Comment 6:**

Lines 1011-1012: What about the role of $NO_3$ chemistry? How is that separated from
aqueous-phase chemistry at night? Was the role of carryover (influence of daytime production)
explored?

*Response:*

*We presumed that the $NO_3^-$ chemistry is probably linked with aqueous-phase chemistry. A*
*previous study reported that the portioning of organic compounds to the particle phase was*
*significantly increased at elevated RH levels (70%) in an urban area dominated by biogenic*

*emissions in Atlanta (Hennigan et al., 2008). From our data, the correlation of ON and LO-OOA in summer nighttime (r=0.76) was stronger than that during daytime (0.53). The concurrent enhancement of the LWC and nitrate functionality from organic nitrate during nighttime demonstrates that the LWC does not inhibit increases in ON concentration, as might be expected if hydrolysis of ON occurred rapidly. This is included in the text on lines 502-505.*

*The importance of carryover of daytime production of water-soluble organic gases (WSOG) and their effect on aq-SOA depends on the expected atmospheric lifetime or uptake reversibility of WSOG (Hodas et al. 2014). This is also true of semi-volatile organic products that partition as a function of temperature. Given that the measurement of either gas-phase water-soluble or semi-volatile organic compounds was not included in this work, it is not possible to comment on this possibility in this dataset.*

**Comment 7:**

Lines 1048-1050: The statement that aerosol water facilities formation of ON needs to be supported. In general, hydrolysis is thought to be a sink of ON. Couldn't this be due to higher organic nitrate yields from $NO_3$ + BVOC chemistry compared to OH + BVOC chemistry?

*Response:*

*We thank the referee for highlighting this fact. The original statement has been rephrased as follows, now on lines 500-505:*

*"This is likely due to the higher ON yields from $NO_3^-$-initiated chemistry involving BVOCs during nighttime compared to hydroxyl radical + BVOCs chemistry during daytime. Additionally, the concurrent enhancement of the LWC and nitrate functionality from organic nitrate during nighttime demonstrates that the LWC likely does not inhibit increases in ON concentration."*

**Comment 8:**

Resolution of the figures needs to be improved (particularly the time series figures). They are currently difficult to read.

*Response:*

*The figures have been updated with high resolution.*

[revised manuscript text omitted]